# A survey of optimal strategy for signature-based drug repositioning and an application to liver cancer

**Chen Yang[1†], Hailin Zhang[1†], Mengnuo Chen[1†], Siying Wang[1†], Ruolan Qian[1], Linmeng Zhang[1], Xiaowen Huang[2], Jun Wang[1], Zhicheng Liu[3], Wenxin Qin[1], Cun Wang[1]\*, Hualian Hang[1]\*, Hui Wang[1]\***

[1]State Key Laboratory of Oncogenes and Related Genes, Department of Liver Surgery & Shanghai Cancer Institute, Renji Hospital, Shanghai Jiao Tong University School of Medicine, Shanghai, China; [2]Key Laboratory of Gastroenterology and Hepatology, Division of Gastroenterology and Hepatology, Renji Hospital, Shanghai Jiao Tong University School of Medicine, Shanghai, China; [3]Hepatic Surgery Center, Tongji Hospital, Tongji Medical College, Huazhong University of Science and Technology, Wuhan, China

**\*For correspondence:**
cwang@shsci.org (CW);
hanghualian@shsmu.edu.cn (HH);
hwang@shsci.org (HW)

[†]These authors contributed equally to this work

**Competing interest:** The authors declare that no competing interests exist.

**Abstract** Pharmacologic perturbation projects, such as Connectivity Map (CMap) and Library of Integrated Network-based Cellular Signatures (LINCS), have produced many perturbed expression data, providing enormous opportunities for computational therapeutic discovery. However, there is no consensus on which methodologies and parameters are the most optimal to conduct such analysis. Aiming to fill this gap, new benchmarking standards were developed to quantitatively evaluate drug retrieval performance. Investigations of potential factors influencing drug retrieval were conducted based on these standards. As a result, we determined an optimal approach for LINCS data-based therapeutic discovery. With this approach, homoharringtonine (HHT) was identified to be a candidate agent with potential therapeutic and preventive effects on liver cancer. The antitumor and antifibrotic activity of HHT was validated experimentally using subcutaneous xenograft tumor model and carbon tetrachloride ($CCL_4$)-induced liver fibrosis model, demonstrating the reliability of the prediction results. In summary, our findings will not only impact the future applications of LINCS data but also offer new opportunities for therapeutic intervention of liver cancer.

## Editor's evaluation

This paper describes a new method and experimental manipulations to identify homoharringtonine as a new potential therapy for liver cancer and the underlying liver disease.

## Introduction

Despite the major advances in drug research and development (R&D), the cost for de novo drug development remains high, ranging from $3 billion to more than $30 billion. Moreover, it usually takes over 10 years to bring a new drug from bench to bedside, reflecting the complex challenges in this area (*Scannell et al., 2012*). Within this context, exploring new indications for existing drugs (drug-centric) or identifying effective drugs for certain diseases (disease-centric) represents an appealing concept, namely 'drug repositioning' (or 'drug repurposing'), which can greatly shorten the gap between preclinical drug research and clinical applications (*Ashburn and Thor, 2004*; *Liu et al., 2013*). Leveraging big data-driven approaches, drug repositioning can be conducted computationally, which

**Figure 1.** Overview of LINCS data-driven therapeutic discovery. The working principle of 'signature reversion'-based computational approach. A disease signature representing discordant expression pattern needs first to be identified (G1, G2, and G3 stand for upregulated genes while G4, G5, and G6 stand for down-regulated genes in disease state). With this signature, pharmacologic perturbation data sets can be queried to find compounds with the ability to reverse disease expression pattern (suppress expression of G1, G2, and G3 and induce expression of G4, G5, and G6). After determining the candidate compounds, experimental and clinical validation are required to translate computational findings to clinical applications. LINCS, Library of Integrated Network-based Cellular Signatures.

The online version of this article includes the following figure supplement(s) for figure 1:

**Figure supplement 1.** A summary of potential factors influencing the accuracy of signature reversion-based computational approach.

**Figure supplement 2.** An overview of compound-induced expression profiles in LINCS.

has the potential to complement traditional therapeutic discovery means and further improve the cost-effectiveness of drug development (*Li et al., 2016*). The most notable data resources supporting the in silico-based therapeutic discovery campaigns would be the Connectivity Map (CMap) (*Lamb et al., 2006*) and its recent extension called Library of Integrated Network-Based Cellular Signatures (LINCS) (*Subramanian et al., 2017*). These two projects have generated large-scale drug-induced gene expression profiles on multiple cancer cell lines under different treatment conditions (CMap Build 2: 3 cell lines, 1309 compounds; LINCS: 77 cell lines, 19,811 compounds), representing a treasure trove for in silico therapeutic exploration (*Musa et al., 2018*). As a 1000-fold scale-up of the original CMap, LINCS contained dramatic increases in both cell line types and perturbations, making it the focus of the present investigation.

The computational drug discovery approach using LINCS (also CMap) data is based upon a basic concept called 'signature reversion' (*Li et al., 2016*). Briefly, compounds with the ability to reverse disease-specific gene expression pattern are considered therapeutic candidates (*Figure 1*). To date, although there have been many successful applications, many problems with this approach remain unsolved (*Chen et al., 2017a*; *Chen et al., 2017b*; *van Noort et al., 2014*). Due to the lack of appropriate benchmarking standards, limited studies have investigated the factors influencing the accuracy of this approach. Therefore, no consensus regarding the implementation details has been reached across current studies. Constructing rational benchmarking standards and developing the best practice approach can facilitate the development of signature reversion approach and help to identify more effective therapeutic strategies for refractory diseases.

Herein, we mainly focused on the disease of liver cancer. As one of the most lethal malignancies worldwide, liver cancer directly accounts for nearly one million deaths each year (*Bray et al., 2018*). Hepatocellular carcinoma (HCC) is the major type of liver cancer, representing approximately 90% of all liver cancer cases (*Llovet et al., 2016*). Although many standard of care therapies, including Lenvatinib (*Kudo et al., 2018*), regorafenib (*Bruix et al., 2017*), cabozantinib (*Abou-Alfa et al., 2018*), ramucirumab (*Zhu et al., 2019*), pembrolizumab (*Finn et al., 2020b*), nivolumab (*El-Khoueiry et al., 2017*), and atezolizumab-bevacizumab (*Finn et al., 2020a*), have been approved for treating HCC in recent years, most of them can yield only marginal survival benefit. Thus, more effective therapeutics treatments for HCC are highly desired. The objectives of the present study were threefold. The first objective was to develop novel benchmarking standards for evaluating drug retrieval performance. The second one was to determine the best practice approach for LINCS data-based signature reversion. For the last objective, we sought to identify novel drug candidates against liver cancer, exploiting the findings from the second objective.

## Results
### Summary of influencing factors and compound experiments in LINCS

Many factors may affect the accuracy of signature-based drug retrieval. We have categorized these factors into three main aspects: acquisition of compound signature (reference signature), generation of disease signature (query signature), and selection of disease-compound matching methods (*Figure 1*, *Figure 1—figure supplement 1*). Although all factors were mentioned and discussed, not all of them were included in the present analyses, considering that some factors have been covered elsewhere and some were challenging to explore due to data and method restrictions. In this study, systematic analyses were carried out to assess the influences of four major factors on signature matching-based drug discovery, including source of cell line, clinical phenotype of query signature, query signature size, and signature matching method.

Since only compound-induced expression data was the focus of this study, we first excluded experiments of other perturbagens, including gene knockdown (or knockout) and gene overexpression manipulations. Subsequently, the distribution of compound profiles was visualized based on their perturbation time, perturbation dose, and cell line used. Most of the measurements were made in the treatment durations of 6 hr (43%) or 24 hr (56.6%), and under the concentrations of 5 μM (21%) or 10 μM (63%) (*Figure 1—figure supplement 2A*). The count distribution of all cell lines in LINCS was also presented in *Figure 1—figure supplement 2B*. Although 71 cell lines were included in LINCS project in total, not all of them were extensively profiled, and only 9 cell lines contained more than 5000 profiles, which, however, account for 77.8% of all compound profiles. There were 2912 compounds shared by these nine cell lines. We further integrated annotation of the most profiled cell lines with treatment duration and concentration information, and illustrated the specific profile numbers of each cell line under the conditions of certain time and dose (*Figure 1—figure supplement 2C*). Unless otherwise indicated, all the following analyses were performed on a fixed perturbation condition of 10 μM for 6 hr. Besides, compound profiles of all top nine cell lines were only utilized when investigating the factor 'Source of cell line.' In other cases, we focused exclusively on the cell line of HepG2, as our main point was to uncover novel therapeutics for liver cancer in this study. A systematic summary of included data sets for analyses was presented in *Supplementary file 1A and B*.

## Compound-induced expression changes are highly cell line-specific

Some previous studies utilized compound profiles from cell lines irrelevant to the disease they are studied for signature reversion prediction. To investigate whether this was a reasonable practice, we conducted following analyses based on LINCS data of the nine most profiled cell lines. First, we visualized the compound profiles in a cosine distance-based two-dimensional t-distributed stochastic neighbor embedding (t-SNE) plot that represented the overall compound perturbation space wherein each dot was equivalent to a unique perturbation and each cell line was color-coded (*Figure 2A*). As shown in the figure, most dots with the same color clustered together, indicating that most of compound-induced gene expression changes tended to be cell-type specific. Intriguingly, dots with different colors in the white region seemed to mix together, suggesting that some compounds might induce similar gene expression changes across cell lines. To figure out which compounds were likely to cause cell-type specific gene expression changes and which tended to induce universal changes independent of cell lines, we calculated the pairwise cosine similarities (L1) among the profiles from the same compounds measured in different cell lines (*Figure 2B*). The cosine similarity measures range from –1 to 1, where higher values indicate increased similarity. The similarity scores (compound-level, L2) of the 2912 unique compounds were determined by calculating the median pairwise cosine similarity values (L1) across the nine cell lines (*Supplementary file 2*). As a result, a high degree of cell-specificity was observed for most compounds, with a median L2 similarity score of 0.078 (*Figure 2C*). Furthermore, we retrieved the mechanism of action (MOA) information and mapped them to the compounds to determine the MOA-level similarity scores (L3). L3 similarity scores were calculated based on the median values of L2 similarity scores of compounds within the same MOA. Results suggested that inhibitors targeting core cellular processes (e.g., cell cycle, RNA transcription, and protein synthesis) tended to induce similar changes across all cell lines, generally in agreement with previous findings (*Figure 2D*, *Supplementary file 2*; *Niepel et al., 2017*; *Subramanian et al., 2017*; *Wang et al., 2018*). We then marked the dots representing the compounds of the top five MOAs in the t-SNE plot. As expected, most of marked dots fell in clusters within the nonspecific region (*Figure 2E*).

Apart from investigating the similarity of perturbed expression profiles at compound level, we further sought to further investigate the cell line pair/cell line-level similarity. Nine cell lines contributed a total of 36 unique cell line pairs. The cell line pair-level perturbed expression similarities (L4) were determined through calculating the median value of similarity scores of all compound pairs between two cell lines, and the corresponding basal expression similarities were computed using Spearman ranked correlation on expression data from CCLE project (*Figure 2F*). The result showed that there was a significant, albeit not very remarkable, association between the perturbed expression similarities (cell line pair-level, L4) and basal expression similarities ($\rho$ =0.344; p=0.040), suggesting that cell lines with similar molecular features were more likely to have consistent gene expression changes upon perturbation (*Figure 2G*). Similarities within the nine cell lines were also explored (cell line-level, L5). Among the nine cell lines we tested, PC3 cell line showed the highest L5 similarity score (median value=0.122) (*Figure 2H*). Notably, the cosine similarity of 0.122 still denoted a weak relationship, which further supported the conclusion that compound-induced gene expression changes were highly cell line-specific.

Among the nine most profiled cell lines, HepG2 was the only one derived from liver. To investigate whether HepG2 was an appropriate cell line model for computational therapeutics discovery for liver cancer or other liver-associated diseases, we calculated the expression correlation between HepG2 and other cell lines (921 CCLE cell lines) or tissues (17,382 normal tissues from GTEx and 9701 tumor tissues from TCGA PanCancer). Compared to other tissue-derived cancer cell lines or normal/tumor tissues, HepG2 exhibited a significantly higher expression correlation with liver cancer cell lines (median correlation coefficient=0.729), normal liver tissues (median correlation coefficient=0.616), and liver cancer tissues (median correlation coefficient=0.631) (*Figure 2—figure supplement 1*). Collectively, we supposed that the use of LINCS-derived HepG2 data was preferable to be limited within liver diseases.

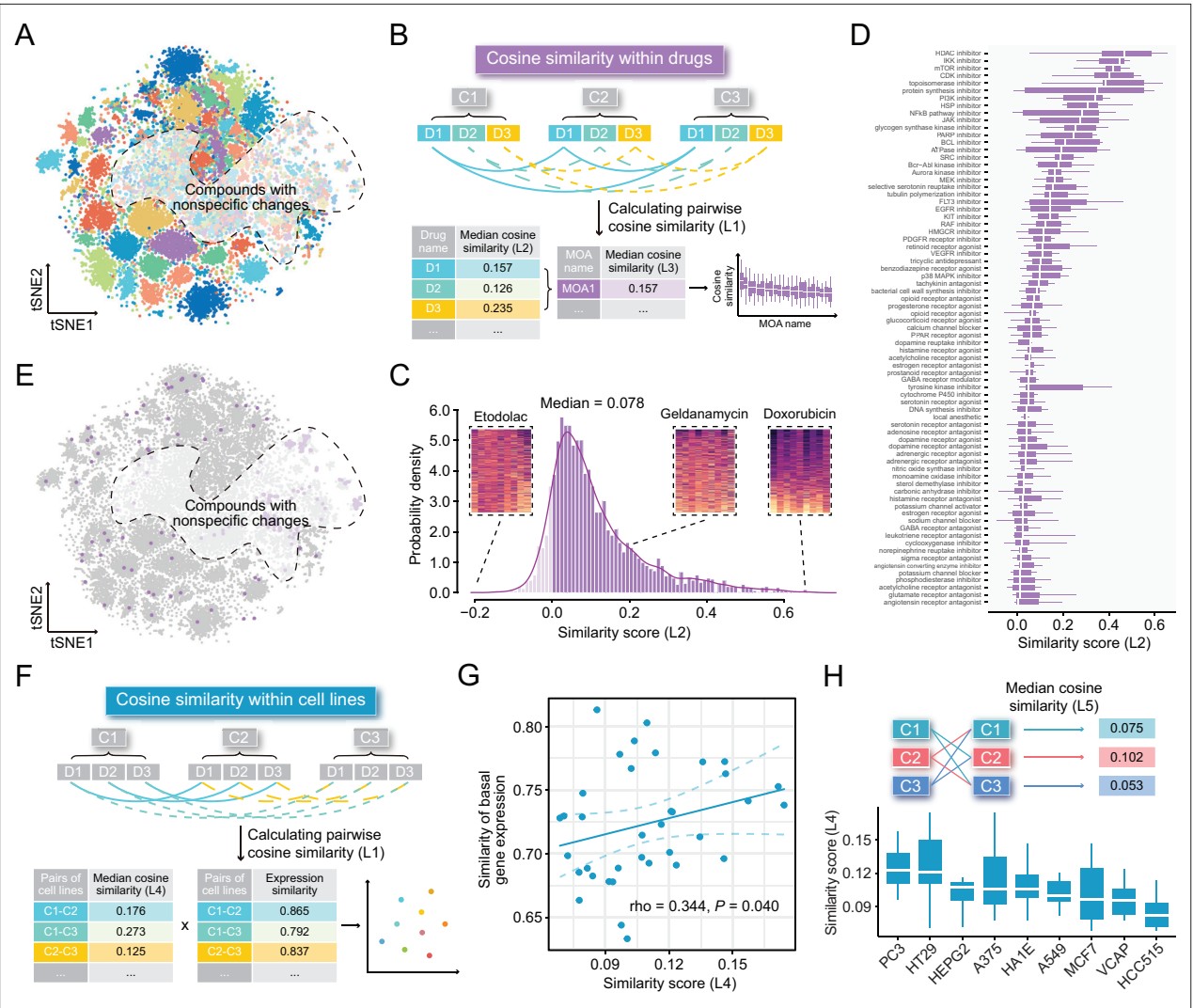

**Figure 2.** Highly cell-type specific compound-induced expression changes. (**A**) Two-dimensional t-SNE projection based on cosine distance between compound signatures. Each dot represents a unique perturbation-induced expression profile, and each color represents one type of cell line. Drug perturbation data was obtained from GSE92742 and GSE70138. (**B**) Schematic diagram displaying the calculation process of compound-level (L2) and MOA-level (L3) similarity scores. (**C**) Distribution of compound-level (L2) cosine similarity scores, which range from –1 (completely opposite pattern) to 1 (perfectly identical pattern). Three examples are presented (left to right: etodolac, geldanamycin, and doxorubicin). (**D**) Illustration of MOA-level (L3) similarities. Only MOAs with more than five compounds included are shown in the figure. (**E**) A t-SNE projection showing the distribution of compounds (indicated by purple dots) in top ranked five MOAs (including HDAC inhibitors, IKK inhibitors, mTOR inhibitors, CDK inhibitors, and topoisomerase inhibitors). (**F**) Schematic diagram displaying the calculation process of cell line pair-level (L4) similarity scores. (**G**) Correlation between basal expression similarities and perturbed expression similarities (L4) of 36 cell line pairs (nine cell lines in total). Statistical significance and correlation coefficient were determined by ranked-based Spearman correlation. (**H**) Schematic view of the calculation of cell line-level (L5) similarity scores (upper) and the presentation of L5 similarity scores of nine cell lines in the boxplot (lower). Data are presented as median±quartiles. MOA, mechanism of action; t-SNE, t-distributed stochastic neighbor embedding.

The online version of this article includes the following figure supplement(s) for figure 2:

**Figure supplement 1.** Correlations between HepG2 cell line and other cancer cell lines or normal/tumor tissues.

## Developing benchmarking standards for evaluating drug retrieval performance

Owing to the lack of benchmarking standards, accurate assessment of retrieval performance of signature matching methods remains challenging. Inspired by previous findings (*Chen et al., 2017a*; *Chen et al., 2017b*; *Cheng et al., 2014*; *Wagner et al., 2015*), we proposed two novel benchmarking standards, namely area under the curve (AUC)-based standard and Kolmogorov-Smirnov (KS)

statistic-based standard. They were built upon different notions and thus were independent of each other, which helped to avoid potential bias introduced by single standard. The corresponding benchmarking data sets were developed mainly based on preclinical/clinical data of liver cancer. Detailed processes of data collection and metrics calculation were described in Materials and methods and visualized in *Figure 3A*.

The development of AUC-based standard was based on the finding that there existed correlation between the reversal potency and treatment efficacy (*Chen et al., 2017a*; *Wagner et al., 2015*). In order to further validate whether this correlation remained significant in other conditions, we retrieved drug response data from CTRP data set in which area under the dose-response curve (AUDRC) values were used as measurements of drug sensitivity, and utilized two different HCC signatures as query signatures to obtain KS-based similarity scores (*Chen et al., 2017a*; *Chen et al., 2017b*). A total of 109 compounds shared by two data sets were selected to conduct correlation analyses. As a result, statistically significant correlation could still be observed between similarity scores and AUDRC values in these scenarios, further proving the reliability of this standard (*Figure 3B*). A benchmark data set was then generated, composed of 117 unique compounds with both LINCS and drug efficacy (IC$_{50}$) data available, which was taken as a basis for the application of AUC-based standard (*Supplementary file 3*). The resultant AUC from this standard was termed as drug retrieval-associated AUC (DR-AUC). Higher DR-AUC value indicated better performance.

As for KS statistic-based standard, we assumed that agents under evaluation in clinical trial for HCC treatment, namely HCC-related agents, might possess an increased reversal capacity (*Chen et al., 2017b*). In other words, HCC agents were more likely to cause negative enrichment in KS test. To verify this hypothesis, we compiled a set of 27 potential HCC agents which were both included in LINCS and under clinical trials for liver cancer treatment. Besides, similarity scores of all compounds tested in HepG2 were also calculated, which were then used as ranked list for KS test. The results of KS test demonstrated that the HCC agent set was indeed negatively enriched (*Figure 3C*). The resultant enrichment scores (ES) here were termed as drug retrieval-associated ES (DR-ES) (*Supplementary file 4*). Of note, in contrast to DR-AUC, lower DR-ES values denoted better performance.

## XSum is the optimal signature matching method for drug retrieval

The two independent benchmarking standards enabled us to quantitatively assess the retrieval performance of different signatures matching methods. Six available methods, including eXtreme Sum (XSum) (*Cheng et al., 2014*), eXtreme Cosine (XCos) (*Cheng et al., 2013*; *Cheng et al., 2014*), eXtreme Pearson (XCor) (*Zhou et al., 2018*), eXtreme Spearman (XSpe) (*Zhou et al., 2018*), KS test (*Lamb et al., 2006*), and the Reverse Gene Expression Score (RGES) (*Chen et al., 2017a*), were included for performance comparison. To minimize technical bias introduced by different query signatures, four HCC signatures with different sizes generated from distinct data sets were utilized for benchmarking (*Supplementary file 5*). Of these, Sig$_{gastro}$ (*Chen et al., 2017b*) and Sig$_{NC}$ (*Chen et al., 2017a*) were directly obtained from previous publications, while Sig$_{LIRI}$ and Sig$_{GSE54236}$ were generated using RNA-seq data from LIRI cohort and microarray data from GSE54236, respectively. A brief summary of the above essential components involved in the evaluation process was presented in *Figure 4A*.

Considering that the performance of eXtreme methods (including XSum, XCos, XCor, and XSpe) may be affected by the number of top genes (topN), we thus calculated the DR-AUC or DR-ES values of each eXtreme methods iteratively, using topN ranging from 50 to 489. In the condition of using Sig$_{LIRI}$ as query signature, both benchmarking standards demonstrated that XSum outperformed other five methods across almost all candidate topNs (*Figure 4B*). Concordantly, when using other three query signatures, XSum also achieved better performance compared with other methods, except in the case of using Sig$_{Gastro}$ as query signature and AUC-based standard for benchmarking, where RGES showed a similar performance with XSum (*Figure 4—figure supplement 1A-C*). Generally, XSum exhibited a consistently excellent performance, independently of the query signature and benchmarking standard (*Supplementary file 6A and B*). In addition, our analyses also demonstrated that the recently developed RGES (a modification of the KS method) was superior to the KS method and might serve as an alternative approach for KS-based connectivity mapping (*Chen et al., 2017a*).

We next sought to find the most appropriate topN value for applying XSum method to achieve the best retrieval performance. Directly selecting the exact topN value where corresponding DR-AUC/

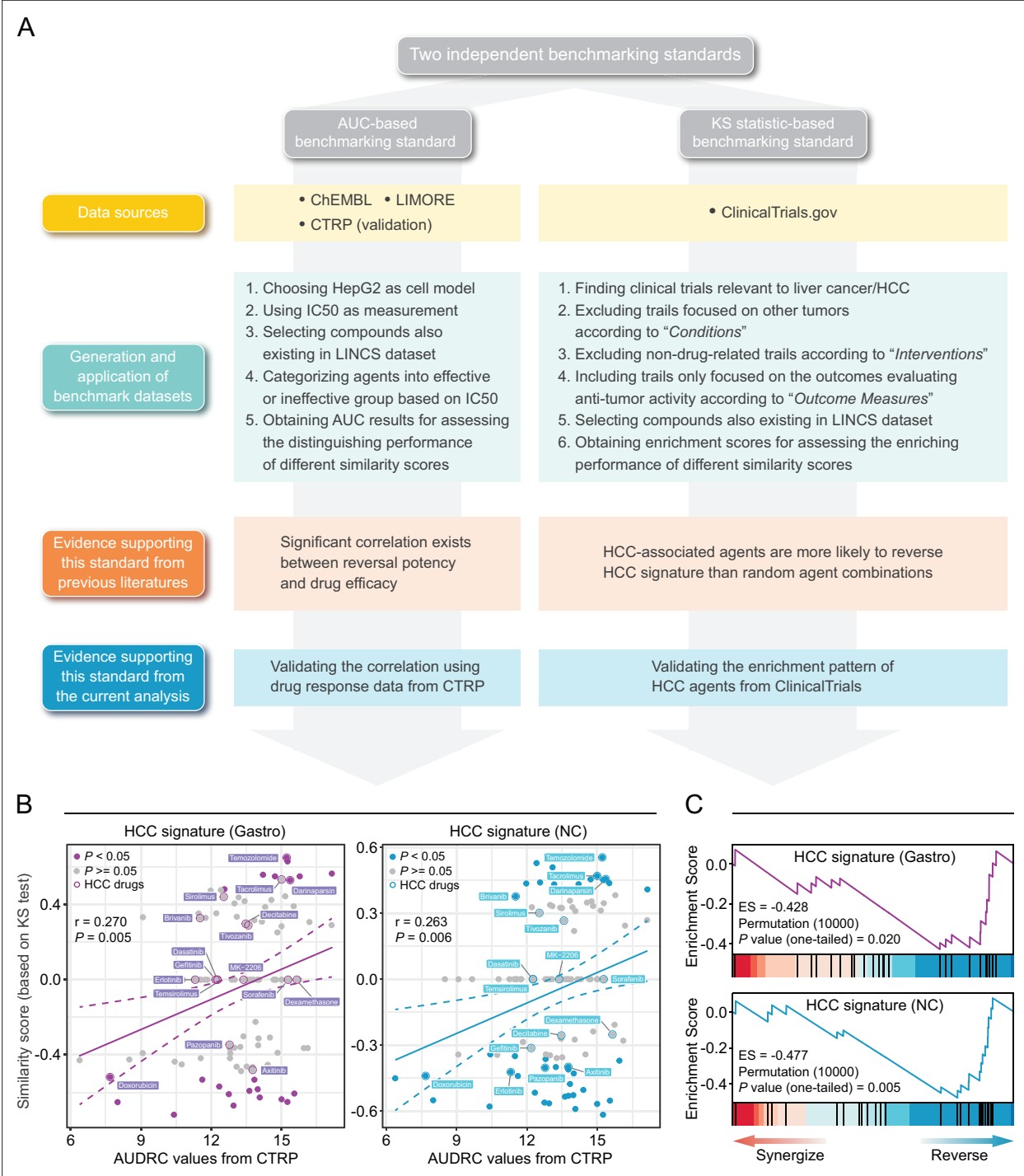

**Figure 3.** Establishment of novel benchmarking standards. (**A**) Flow chart of the data collection and hypothesis validation for the AUC-based (left) and KS statistic-based (right) benchmarking standards. (**B**) Correlation between drug efficacy (AUDRC values) and reversal potency (KS-based similarity scores). Two previously published query signatures, including Sig$_{gastro}$ (left) and Sig$_{NC}$ (right), were utilized to calculate similarity scores. Drug response data was achieved from CTRP data set. Note that lower similarity scores indicate higher reversal potency and lower AUDRC values imply greater drug sensitivity. Color toward gray indicates no statistical significance determined by KS test. (**C**) Reversal potency of HCC agents demonstrated by enrichment analysis. Sig$_{gastro}$ (upper) and Sig$_{NC}$ (lower) were used to compute similarity scores. AUC, area under the curve; HCC, hepatocellular carcinoma; KS, Kolmogorov-Smirnov.

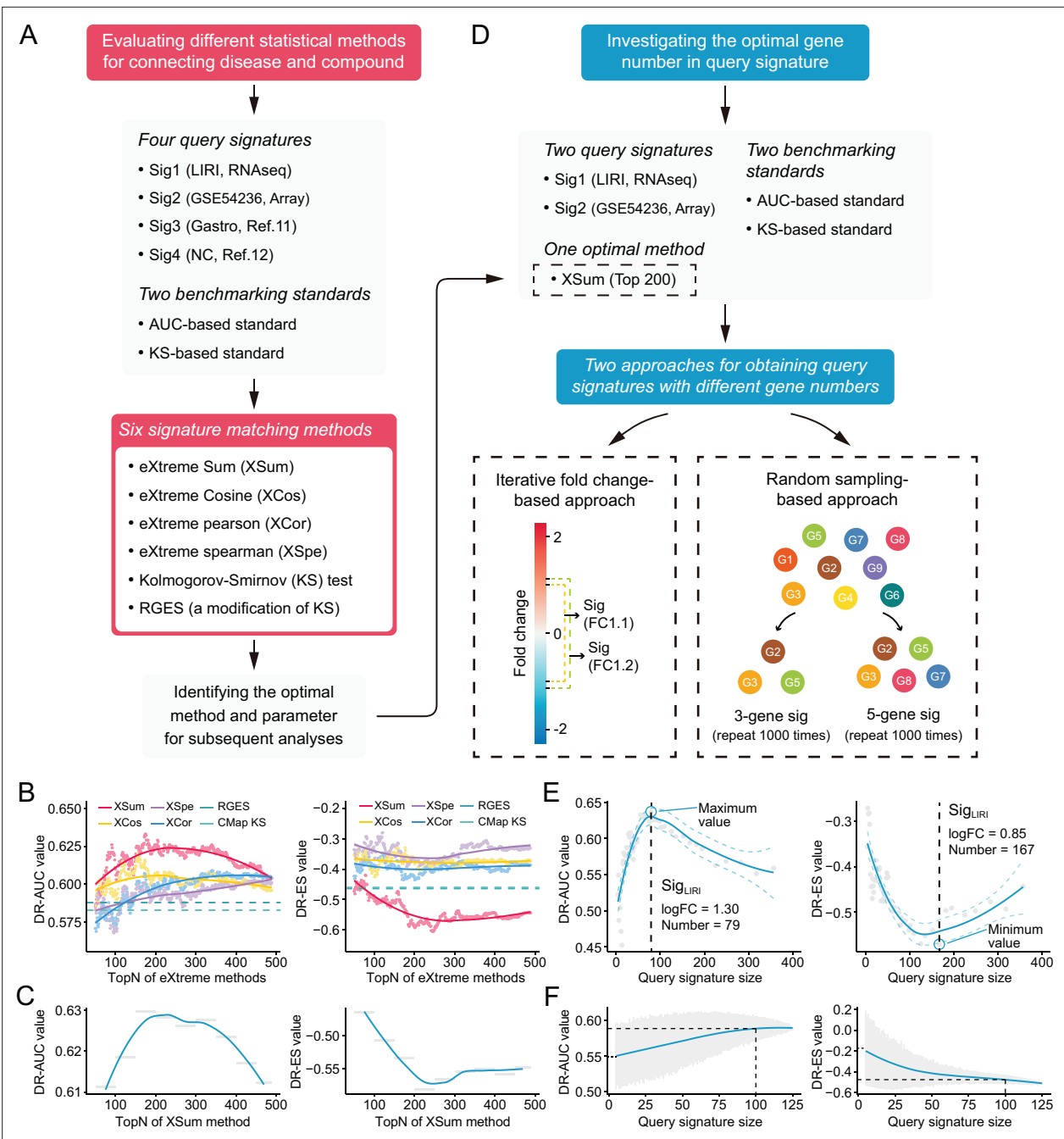

**Figure 4.** Benchmarking different methodologies and parameters. (**A**) Diagram summarizing the workflow and the important components involved in the evaluation process of drug retrieval performance of six different signature matching methods. (**B**) Retrieval performance of six matching methods evaluated by AUC-based benchmarking standard (left) and KS statistic-based benchmarking standard (right). Query signature was generated based on LIRI cohort. (**C**) Visualization of AUC-based (left) and KS statistic-based (right) performance measurements of XSum method on standardized data for discerning the optimal operating parameter. (**D**) Diagram summarizing the workflow and the important components associated with the investigation process of the optimal query signature size. (**E**) Relationship between query signature size determined by iterative fold change-based approach and retrieval performance evaluated by AUC-based standard (left) and KS statistic-based standard (right). (**F**) Relationship between query signature size determined by random sampling-based approach and retrieval performance evaluated by AUC-based standard (left) and KS statistic-based standard (right). LOESS polynomial regression analysis was performed for curve fitting. AUC, area under the curve; KS, Kolmogorov-Smirnov.

The online version of this article includes the following figure supplement(s) for figure 4:

**Figure supplement 1.** Benchmarking methodologies and parameters in the conditions of using different query signatures.

**Figure supplement 2.** The influences of query signature size on retrieval performance.

DR-ES reached their maximum/minimum might cause bias and could be prone to overfitting. Given the continuous trait of candidate topNs, we chose to divide them into several smaller bins, typically 50 topN values in each bin. The DR-AUC/DR-ES of a given bin was defined as the mean DR-AUC/DR-ES values within this bin, which could decrease potential influences brought about by outliers. With this normalization approach, relatively consistent results across varying conditions were obtained. The optimal window with the best performance was either 'top150–200' or 'top200–250' (*Supplementary file 6C*). Besides, we also observed a biphasic pattern of fitting curves, with the inflection points appearing where topNs were around 200 (*Figure 4C*, *Figure 4—figure supplement 1D-F*). Collectively, we supposed that topN of 200 could serve as a rough guide.

## A query signature size of 100 is applicable for drug retrieval

Next, we intended to further discern the optimal query signature size. Considering that $Sig_{gastro}$ ($N_{gene}$=44) and $Sig_{NC}$ ($N_{gene}$=73) had fixed and relatively small signature sizes, only signatures generated from LIRI and GSE54236 cohorts were utilized for the following investigation. We adopted two complementary approaches: (i) iterative fold change-based and (ii) random sampling-based approaches, to obtain query signatures with varying sizes (*Figure 4D*). The iterative fold change-based approach could create a number of signatures with discontinuous sizes through setting iterative threshold values of fold changes. The exact sizes of optimal signatures identified by this approach varied substantially (including 55, 79, 140, and 167). Despite this, similar trends of biphasic pattern with inflection points at around 100 under different conditions could still be observed (*Figure 4E*, *Figure 4—figure supplement 2A*). The approach based on random sampling was adopted as a complement. The results showed that, as the signature sizes increased, the DR-AUC/DR-ES values also increased/decreased and eventually converged when the signature size was more than 100 (*Figure 4F*, *Figure 4—figure supplement 2B*). Accordingly, we considered that a signature size of 100 could be selected as a good compromise. This conclusion remained valid in the conditions when other topN values were applied, such as 100 and 400 (results not shown).

## A good query signature should comprehensively reflect the clinical characteristics of corresponding disease

Many previous studies chose to compare normal versus diseased states to define disease signatures. However, signatures that are generated based on other clinical phenotypes, such as prognosis and metastasis, can also be used to query LINCS. Aiming to figure out whether this factor could also affect the performance of drug retrieval, we designed a forward and a backward strategy (*Figure 5A*). The application of forward strategy was based on two types of signatures, general signatures (representing discordant expression pattern between normal and tumor tissues) and prognostic signatures (associated with survival outcomes). We compared the above two signature phenotypes across varied signature sizes. Unfortunately, the results under different data sets and benchmarking standards were highly inconsistent. This strategy thus failed to provide a definitive conclusion (*Figure 5—figure supplement 1A and B*).

Opposed to the forward strategy, backward strategy started from creating a collection of 10,000 random signatures, followed by determining the optimal signature for clinical implication evaluation. The optimal random signature was determined according to both benchmarking standards. Exploring the clinical values of this signature might reveal some necessary features possessed by a 'good' query signature (*Figure 5B*). A comprehensive clinical evaluation on the optimal signature was carried out based on five RNA-seq and five microarray clinical cohorts from three perspectives. First, the ability of this signature to distinguish tumors from non-tumors was investigated. Briefly, we extracted the first principal components (PC1) of this signature to represent its overall expression pattern. AUC was used here as a measurement of the classification capability. The results showed that more than 0.90 of AUC can be obtained in seven out of the eight cohorts (87.5%), indicating that the ability to discern the difference between diseased and normal states might be an indispensable property for achieving good retrieval performance (*Figure 5C*). Next, we intended to find out whether the optimal signature should be a prognostic indicator. Cox regression analyses were conducted to investigate the association between the signature expression (PC1) and clinical outcome. As a result, significant prognostic implications of the optimal signature could be observed in six out of the eight cohorts (75%), suggesting that prognostic significance was also a necessary characteristic (*Figure 5D*). At last,

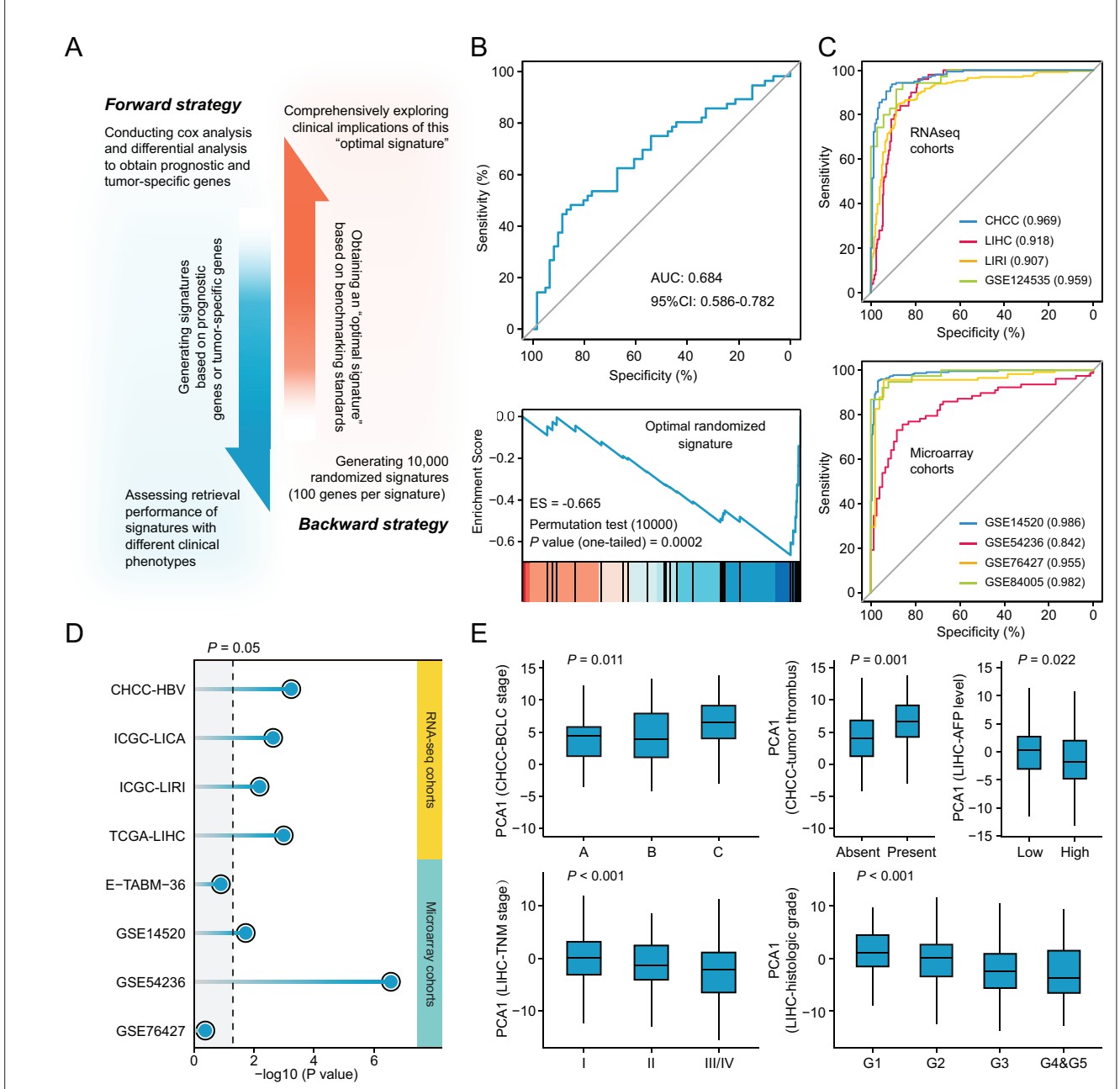

**Figure 5.** Necessary properties of a good query signatures. (**A**) Schematic illustration of forward and backward strategy adopted to investigate whether the factor associated with clinical phenotype of query signature can affect computational therapeutic discovery. (**B**) The DR-AUC value and DR-ES value of the optimal randomized signature showed by ROC curve (upper) and enrichment plot (lower). (**C**) The association between the optimal signature and the clinical phenotype of discordant expression pattern suggested by ROC curves based on RNA sequencing cohorts (upper) and Microarray cohorts (lower). (**D**) The association between the optimal signature and the clinical phenotype of prognosis. Color toward gray indicates no statistical significance. (**E**) The association between the optimal signature and multiple clinical features, including BCLC and TNM stage, tumor thrombus, AFP level, and histologic grade. Data are presented as median±quartiles. N≥100. Statistical significance of difference between groups was determined using either Kruskal-Wallis or Wilcoxon sum rank tests.

The online version of this article includes the following figure supplement(s) for figure 5:

**Figure supplement 1.** The influences of query signature phenotype on retrieval performance.

the association between signature expression and other clinical features was explored. Considering that CHCC and LIHC cohorts held the most abundant clinical information, corresponding analyses were thus conducted on these two cohorts. The results showed that there was a significant correlation between the optimal signature and multiple clinical features, including BCLC stage (p=0.011), tumor thrombus (p=0.001), AFP level (p=0.022), TNM stage (p<0.001), and histologic grade (p<0.001) (*Figure 5E*). Accordingly, we concluded that a good query signature should possess the ability to comprehensively recapitulate the clinical features of corresponding disease, rather than only reflect the disease characteristic from single perspective.

## Generation of novel liver cancer signature

The conclusions from the above analyses were then applied to establish a signature representing liver cancer initiation and development, which could be utilized to query compounds with potential therapeutic as well as preventive effects against liver cancer. The generation of this evolution-associated signature was based on the concept that the initiation and progression of liver cancer was a step-wise process with gradually acquired advantageous biological capabilities (*Figure 6A*). Therefore, conceptually, antagonizing genes that were most related to these stages could be a potential therapeutic strategy. Through implementing random forests algorithm on GSE89377 cohort, preliminary screening was performed to include stage-associated genes, where genes with greater predictive power were selected for further analysis. This screening yielded a total of 6017 stage-associated genes (23.9%), of which 309 were landmark genes (*Figure 6B*). Next, we conducted weighted gene co-expression network analysis (WGCNA) to obtain co-expressed modules with diverse expression patterns (*Figure 6—figure supplement 1A*). Seven gene modules were discerned by WGCNA analysis (*Figure 6—figure supplement 1B and C*), and two of them, which we termed the 'ascending' module (N=1738) and the 'descending' module (N=350) for their greatest relevance to stages and patterns of linear evolution from normal to cancer, were retained for further analyses (*Figure 6C and D*). Biological processes associated with genes in these two modules were investigated. We found that the 'ascending' module was closely associated with proliferation (*Figure 6C*), while the 'descending' module was enriched in several different types of processes (*Figure 6D*). There were 159 genes in common between these two modules and landmarks. Based on the aforementioned recommendation of query signature size, we sought to further reduce the size of 159–100. This procedure was carried out using HCC occurrence-related clinical and molecular data from GSE15654 cohort. In brief, 10,000 random signatures, each containing 100 genes, were generated based on the 159-gene panel. The one which had the most significant association with HCC occurrence was considered as the optimal query signature (*Figure 6—figure supplement 2A and B*). This analysis yielded a signature comprised of 82 ascending genes and 18 descending genes, which was then named as $Sig_{evo}$ (*Supplementary file 7*). The linear evolution pattern of $Sig_{evo}$ remained present in training (*Figure 6—figure supplement 2C*) as well as an independent validation cohort (*Figure 6—figure supplement 2D*).

As previously discussed, a good query signature should reflect the clinical features of corresponding disease comprehensively. Therefore, we systematically surveyed the association between $Sig_{evo}$ and the clinical phenotypes of precancerous/cancerous liver lesions using clinical and experimental data from both human and animal data sets. First, based on clinical cohorts of HCC, we demonstrated that $Sig_{evo}$ had a remarkable capability for distinguishing tumors from non-tumors, with a median AUC of 0.972 in all eight cohorts (*Figure 6E*). Besides, this signature also held great prognostic power in HCC, as indicated by the results of Cox analyses (*Figure 6F*). Next, in view of the crucial role of fibrosis in driving hepatocarcinogenesis, further investigation was performed to validate its relevance to fibrosis-related phenotype. The result suggested that $Sig_{evo}$ could also effectively differentiate between mild (S0/S1) and severe (S3/S4) fibrosis (*Figure 6G*). Additionally, we collected four experimental data sets, including two carbon tetrachloride ($CCl_4$)-treated mouse data sets and two diethylnitrosamine (DEN)-treated rat data sets, to assess the enrichment levels of $Sig_{evo}$ in mouse and rat fibrosis models. It could be observed that ascending genes in $Sig_{evo}$ were significantly enriched in both $CCl_4$-treated (GSE27640) and DEN-treated (GSE19057) liver tissues (*Figure 6H and I*). However, descending genes did not exhibit any significant enrichment pattern in all included data sets, possibly due to the limited gene number (*Figure 6H, I*). Notably, the expression profiles in GSE63726 were derived from non-parenchymal cell fractions which had abundant hepatic stellate cells (HSCs), and thus the significant enrichment could provide the evidence that this signature might reflect the molecular feature of HSC

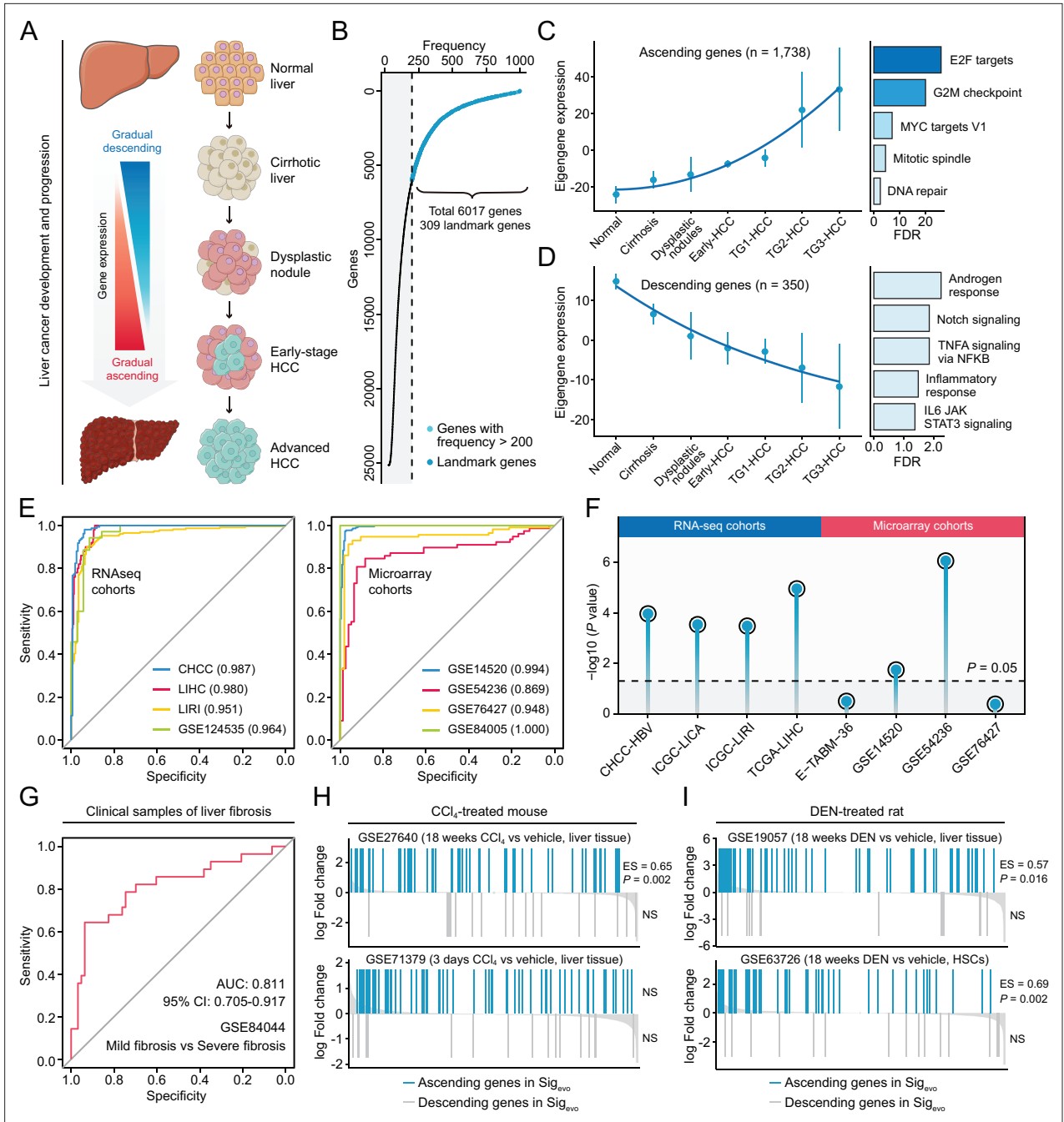

**Figure 6.** Development of a novel signature representing the initiation and progression of liver cancer. (**A**) Schematic of the stepwise process of liver cancer initiation and progression. (**B**) Preliminary screening of developmental stage-associated genes by random forests algorithm based on GSE89377. (**C**) The expression pattern of the 'ascending' module discerned by WGCNA analysis (left) and the enriched biological processes determined by hypergeometric test (right). (**D**) The expression pattern of the 'descending' module (left) and the enriched biological processes (right). (**E**) The performance evaluation of the Sig$_{evo}$ for discerning the difference between tumor and normal tissues based on RNA sequencing cohorts (left) and microarray cohorts (right). (**F**) The association between the Sig$_{evo}$ and the clinical phenotype of prognosis. Color toward gray indicates no statistical significance. (**G**) The association between the Sig$_{evo}$ and fibrosis-related phenotype suggested by ROC curve. (**H**) The association between Sig$_{evo}$ and CCl4-induced expression changes in liver tissues of mice. The enrichment scores and statistical significance were determined by gene set enrichment analysis. (**I**) The association between Sig$_{evo}$ and DEN-induced expression changes in liver tissues of rats. WGCNA, weighted gene co-expression network analysis.

The online version of this article includes the following figure supplement(s) for figure 6:

**Figure supplement 1.** Weighted gene co-expression network analysis (WGCNA).

**Figure supplement 2.** Identification and validation of the novel query signature.

activation (*Figure 6I*). In summary, the Sig$_{evo}$ fully complied with the criteria of a good query signature and was then employed for querying LINCS.

Using the optimal method (XSum) and a compromising parameter (topN=200), we matched Sig$_{evo}$ with HepG2-derived compound signatures in LINCS and obtained the similarity scores of all compounds (lower scores implied higher reversal potency and greater potential for application). After excluding preclinical agents or agents withdrawn from the market, 793 agents remained (*Corsello et al., 2017*). These agents were then considered repositioning candidates (*Supplementary file 8*). Interestingly, some agents which were previously proved to have chemopreventive effects, including erlotinib (*Fuchs et al., 2014*), caffeine (*Hoshida et al., 2012*), and fasudil (*Nakagawa et al., 2016*), dominated relatively high rankings on the list (*Figure 7A*). Besides, anti-HCC agents were also found to be enriched significantly in compounds with reversal potency (*Figure 7B*). These findings collectively supported the reliability of the prediction results.

## Homoharringtonine is a candidate anti-liver agent

According to the computational results, homoharringtonine (HHT) (*Figure 7C*), a protein synthesis inhibitor targeting RPL3, had the highest reversal potency among 793 repositioning candidates (*Tujebajeva et al., 1989*). To prove that the reversal effect of HHT is not cell type- or concertation-dependent, we generated HHT-perturbed expression data using five different liver cancer cell lines (Hep3B, HepG2, Huh6, Huh7, and PLC) and four different concentrations (0.1 μM, 0.5 μM, 1 μM, and 10 μM). HHT with a fixed concentration of 10 μM (a standard concentration in CMap and LINCS) was used to treat different cell lines and a single cell line HepG2 (a cell line used in LINCS) was perturbed by HHT with varying concentrations (*Figure 7—figure supplement 1A*). HHT signatures were obtained through calculating the fold changes of HHT-treated samples to control samples. Subsequently, GSEA was conducted against different HHT signatures, taking ascending and descending genes in Sig$_{evo}$ as query gene sets separately. The results indicated that the ascending genes tended to enrich in HHT-induced downregulated genes (ES<0), while descending genes appeared to be more associated with HHT-induced upregulated genes (ES>0), suggesting that the ability of HHT to reverse the Sig$_{evo}$ was independent of cell type and treatment concentration (*Figure 7—figure supplement 1B and C*, *Figure 7—source data 1*, GSE193897).

As the drug target of HHT, RPL3 was characterized for its clinical and biological implications. Comprehensive comparisons of the expression of RPL3 between tumor and non-tumor tissues were conducted using seven clinical cohorts with available expression profiles of both tumor and non-tumor tissues. The results showed that RPL3 had higher expression levels in tumor compared with non-tumor tissues in more than half the clinical cohorts (57.1%) (*Figure 7—figure supplement 2A*). The increase of protein expression of RPL3 could also be observed in tumor tissues (*Figure 7—figure supplement 2B*), as shown by immunohistochemical images from the Human Protein Atlas (*Uhlén et al., 2015*). Higher expression of RPL3 also indicated worse prognosis (*Figure 7—figure supplement 2C*). In addition, leveraging CRISPR-based screening data from Project Achilles (*Meyers et al., 2017*), we found that RPL3 was essential for maintaining the survival and growth of all liver cancer cell lines (*Figure 7—figure supplement 2D, E*). Above results demonstrated the rationality of RPL3 inhibition for treating liver cancer.

## Homoharringtonine has a remarkable therapeutic effect against liver cancer

To investigate the in vitro anti-tumor activity of HHT against liver cancer, we analyzed the drug response data of HHT from PRISM data set (*Corsello et al., 2020*). It could be observed that HHT had a lower distribution of IC$_{50}$ values across 482 PRISM cell lines compared with molecular-targeted agents and non-oncology agents (*Figure 7D*). Of note, in liver cancer cell lines, HHT exhibited a powerful tumor suppressor activity with a median IC$_{50}$ value of 0.408 μM, which was numerically lower than that of other three Food and Drug Administration (FDA)-approved HCC agents (lenvatinib: 0.617 μM; regorafenib: 2.009 μM; sorafenib: 3.348 μM) (*Figure 7E*). The in vitro anti-tumor activity of HHT was corroborated by the long-term cell proliferation assay (*Figure 7F*) and short-term IncuCyte real-time assay (*Figure 7—figure supplement 3*). In addition, in vivo efficacy of HHT was also evaluated using subcutaneous xenograft model of MHCC97H cell line. The result demonstrated that HHT could significantly inhibit the growth of xenograft tumors (*Figure 7G and H*), with limited

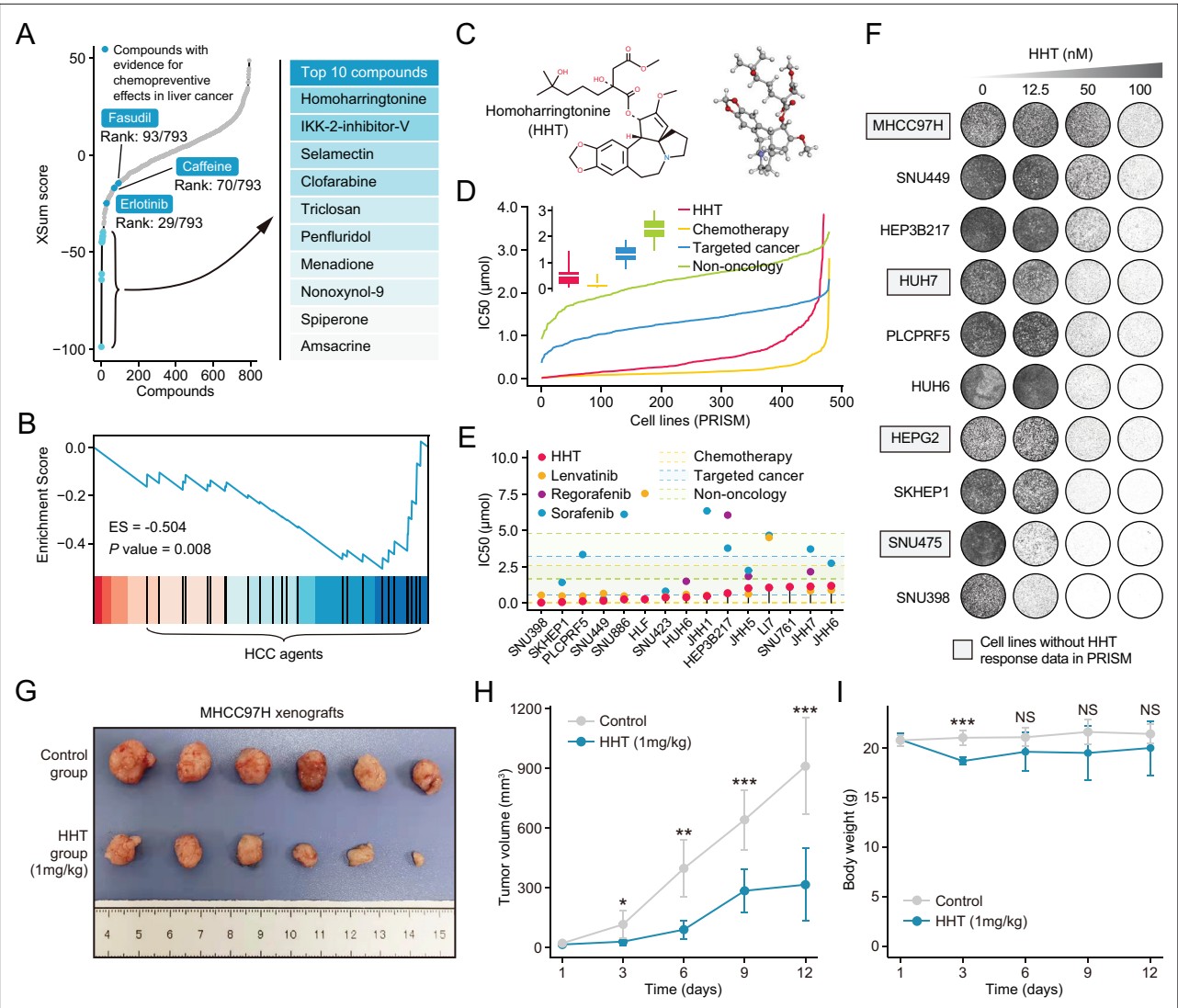

**Figure 7.** Homoharringtonine (HHT) has significant tumor killing activity both in vitro and in vivo. (**A**) Results of best practice approach-based computational dr_evo as query signature. Top ranked 10 compounds with highest reversal potency were illustrated in the right panel. (**B**) Enrichment of HCC agents in compounds with reversal potency (XSum score<0). Statistical significance was determined based on the null distribution formed by 10,000 permutations. (**C**) 2D (left) and 3D (right) chemical structure of HHT. (**D**) Comparison of distribution of compound activity between HHT and three different drug categories, including chemotherapy (N=45 compounds), targeted cancer agents (N=419 compounds), and non-oncology (N=362 compounds). The IC$_{50}$ values (from PRISM data set) of each drug category in each cell line (N=482) were determined through calculating the median IC$_{50}$ value across all the compounds in this category. Data are presented as median±quartiles, N≥100. (**E**) The drug sensitivity data of HHT (achieved from PRISM data set) across liver cancer cell lines. The drug sensitivities of two HCC agents in the first-line (sorafenib and lenvatinib) and one HCC agent in the second-line (regorafenib) were also presented for comparison. Areas with different colors denote the interquartile range of median IC$_{50}$ values of compounds within different drug categories. (**F**) Long-term cell proliferation assay for testing the anti-tumor activity of HHT across 10 liver cancer cell lines. Of these, four cell lines have not been profiled by PRISM for the sensitivity to HHT. (**G**) Macroscopic image of tumors harvested from xenograft mice treated with vehicle (upper) and HHT (lower). (**H**) Longitudinal tumor volume progression of subcutaneous MHCC97H xenograft tumors treated with vehicle (N=6) and HHT (N=6). The statistical significance of difference between groups was determined using Student's t-test. Data are represented as mean ± SD. (**I**) Body weight changes of mice in control (N=6) and HHT-treated (N=6) groups. Statistical significance was determined using Student's t-test. Data are represented as mean ± SD. *p<0.05, **p<0.01, ***p<0.001. NS, not significant. HCC, hepatocellular carcinoma.

The online version of this article includes the following source data and figure supplement(s) for figure 7:

**Source data 1.** Drug-induced expression changes across different cell lines as well as different concentrations.

**Figure supplement 1.** Reversal effect of HHT on Sig_evo across different conditions.

**Figure supplement 2.** Clinical and biological characterization of RPL3 in liver cancer.

**Figure supplement 3.** The effect of HHT on cell proliferation across 10 liver cancer cell lines.

*Figure 7 continued on next page*

*Figure 7 continued*

**Figure supplement 4.** Summary of the anti-tumor effect of HHT-sorafenib combination across 10 liver cancer cell lines.

**Figure supplement 5.** Detailed information of the anti-tumor effect of HHT-sorafenib combination across 10 liver cancer cell lines.

drug-related toxicity (*Figure 7I*). Since co-administration of HHT with other approved agents was more likely to have clinical significance, we also interrogated whether HHT could augment the tumor-killing effect of sorafenib. Three different statistical models were adopted for synergy estimation. The results suggested that HHT could indeed synergize with sorafenib in many conditions, albeit not very remarkable in general (*Figure 7—figure supplements 4 and 5*).

## Homoharringtonine treatment can alleviate liver fibrosis both in vivo and in vitro

Liver fibrosis occurs when the liver tissue is repeatedly and continuously injured, which is a crucial risk factor for hepatocarcinogenesis (*O'Rourke et al., 2018*). Since we have proved that $Sig_{evo}$ was associated with liver fibrosis using clinical and animal-derived data, it could be postulated that HHT also had the potential to alleviate liver fibrosis. The antifibrotic effect of HHT was first assessed using carbon tetrachloride ($CCL_4$)-induced mouse liver fibrosis model (*Figure 8A*). The results suggested that HHT could significantly reduce Ishak scores and positive area of Sirius Red staining compared to vehicle controls (*Figure 8B and C*). Besides, HHT treatment could also lead to significant reduction of serum levels of alanine transaminase (ALT) and aspartate transaminase (AST) (*Figure 8D*). These observations demonstrated that HHT can impede fibrosis development and partially rescued hepatic function in $CCL_4$-induced mouse model. The activation of HSCs is one of the key steps in fibrosis development (*Zhou et al., 2014*). To determine whether HHT could inhibit the activation of HSCs, in vitro experiments based on TGF-β1-activated human HSC line LX-2 were further conducted. LX-2 cells were treated with vehicle or HHT (1 μM and 5 μM) for 6 hr, followed by RNA-seq for quantifying HHT-induced expression changes. Nine fibrotic genes from previous publications were collected; high expression level of these genes represented the activation status of HSCs. After HHT treatments, the expression level of almost all fibrotic genes was downregulated (*Figure 8E*, *Figure 8—figure supplement 1*, *Figure 8—source data 1*, GSE180243). The downregulated tendency of the two most critical genes which encoded collagen I and α-SMA were further corroborated by the quantitative real-time PCR (*Figure 8—figure supplement 2A*). Additionally, the protein-level expression of collagen I and α-SMA was also detected using western blot (*Figure 8—figure supplement 2B*, *Figure 8—source data 2*) and immunofluorescence (*Figure 8—figure supplement 2C*). The results showed that HHT could inhibit the protein expression of collagen I and α-SMA as well. Taken collectively, HHT can inhibit the progression of liver fibrosis via suppressing HSC activation and thus may have certain preventive effects on liver cancer.

## Discussion

In recent years, the explosive growth of pharmacogenomic data enables the development of computational drug discovery and repositioning, leading to many remarkable findings of novel therapeutics (*Kong et al., 2020*; *Stathias et al., 2018*; *Yang et al., 2021*). Owing to the success of CMap and LINCS projects (*Lamb et al., 2006*; *Subramanian et al., 2017*), signature reversion-based computational drug discovery approach has been extensively used (*Chen et al., 2017a*; *Chen et al., 2017b*; *Dudley et al., 2011*; *van Noort et al., 2014*; *Wei et al., 2006*). However, lack of suitable benchmarking standards for evaluating drug repositioning performance limits further improvement of this approach. Some studies proposed that the benchmarks assessing drug-drug similarity, such as the anatomical therapeutic chemical (ATC) system, could be taken as alternative standards to indirectly determine the optimal methodologies and parameters of computational repositioning (*Cheng et al., 2013*; *Zhou et al., 2018*). However, considering the great difference between the two situations, developing tailored benchmarking standards for assessing disease-drug similarity would be more desirable (*Cheng et al., 2014*). In this study, we proposed two novel benchmarking standards, AUC-based and KS statistic-based standards. Despite being mutually independent, the evaluation results of the two standards were highly consistent, demonstrating their rationality and robustness.

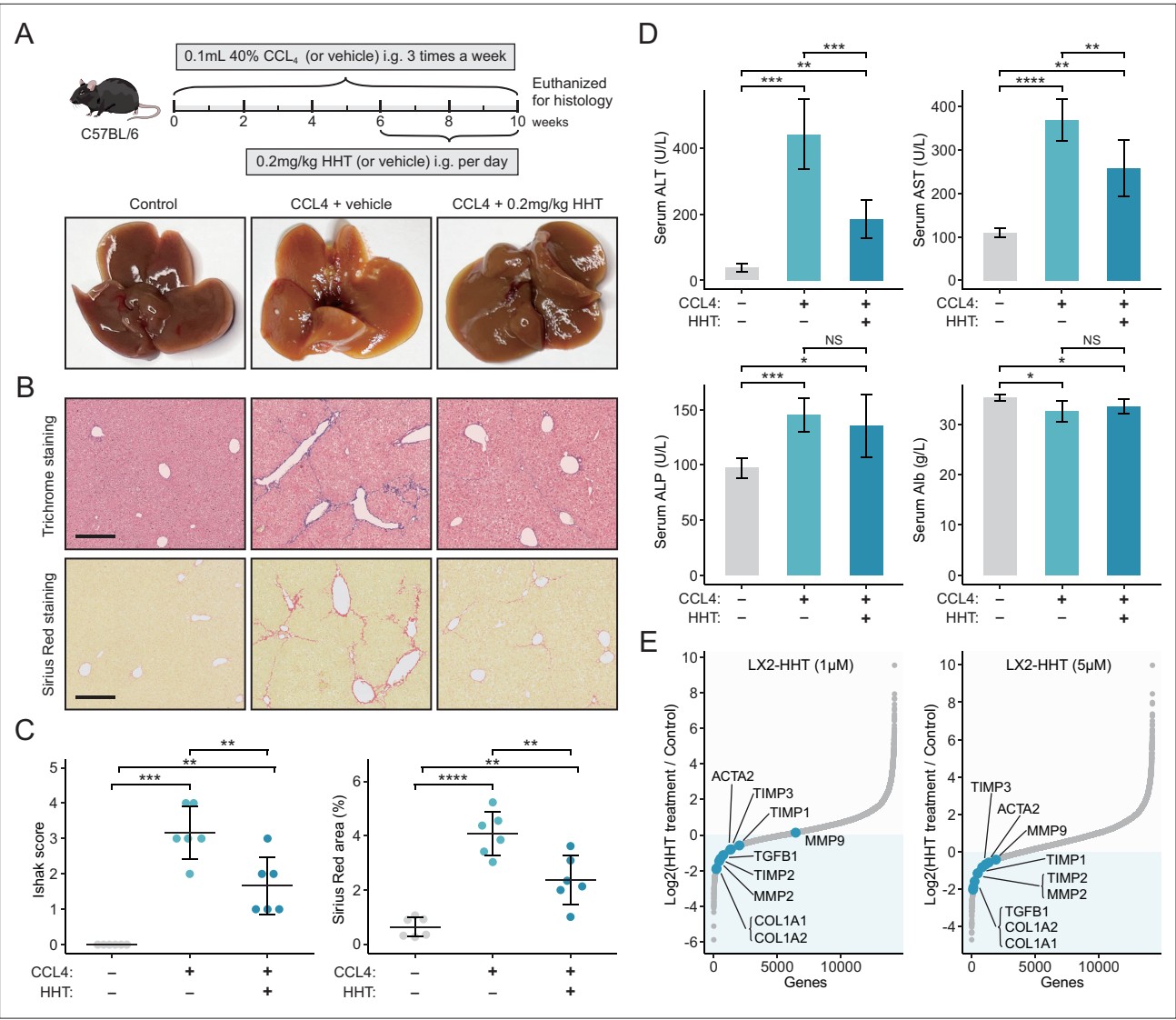

**Figure 8.** HHT has significant in vivo anti-fibrotic effects. (**A**) Schematic diagram (upper) of the experimental design for validating the anti-fibrotic ability of HHT and representative photographs (lower) of the livers harvested from different groups at the time of sacrifice. (**B**) Representative images of Masson's trichrome staining and Sirius Red staining of liver tissues from different groups (scale bars: 250 μm). (**C**) Comparisons of Ishak scores (left) and Sirius Red-based collagen quantification (right) between different groups. Statistical significance was determined using one-way ANOVA followed by Tukey multiple comparison test. Data are represented as mean ± SD (N=6 in each group). (**D**) Comparisons of serum levels of ALT, AST, ALP, and Alb between different groups. Statistical significance was determined using one-way ANOVA followed by Tukey multiple comparison test. Data are represented as mean ± SD (N=6 in each group). (**E**) Differential expression of nine fibrosis-associated genes between HHT-treated and HHT-untreated LX-2 cells. *p<0.05, **p<0.01, ***p<0.001, ****p<0.0001. HHT, homoharringtonine; NS, not significant.

The online version of this article includes the following source data and figure supplement(s) for figure 8:

**Source data 1.** Sequencing results of HHT-treated LX2 cells.

**Source data 2.** Raw unedited plots.

**Figure supplement 1.** Comparison of the expression of nine fibrosis-associated genes between control LX2 and HHT-treated LX2.

**Figure supplement 2.** In vitro anti-fibrotic effects of HHT.

These two standards enable the establishment of a standardized procedure for performing more effective signature-based drug prediction. We first determined that using reference signatures from one of the most relevant cell lines with the disease of interest instead of from a non-touchstone cell line or aggregation-based consensus results was a preferable option to exploit LINCS data. Next, XSum was identified as an optimal method for matching compound and disease signatures.

Interestingly, a prior study that made a comparison of drug retrieval performance between XSum, XCos, and KS methods using a totally different benchmarking standard from ours also come to the same conclusion (*Cheng et al., 2014*). Furthermore, we also uncovered an appropriate parameter of XSum (topN=200), which lacked guidance previously.

Most of the current investigations and methodological developments were focused on reference signatures and signature matching methods. By contrast, relatively limited efforts have been made to standardize the generation of query signatures (*Chan et al., 2019*; *Wen et al., 2016*; *Wen et al., 2015*). In this study, two potential factors, signature phenotypes and signature size, were systematically analyzed. Through adopting two independent approaches, an appropriate query signature size of 100 was determined. However, prior studies considered a reduced number of 50 as the optimal size of query signatures (*Chen et al., 2017a*; *Parkkinen and Kaski, 2014*). It is reasonable to speculate that the utility of different signature matching methods (XSum in this study and KS-based methods in other studies) and also the different benchmarking standards may be responsible for the discrepancy. Next, we determined that a good query signature should hold the ability to comprehensively characterize the clinical features of corresponding disease. This finding seemed to be reasonable since disease was highly likely to be underrepresented when the query signature was generated based on a single clinical phenotype.

Based on these findings, we summarized the best practice approach for LINCS-based drug prediction. An application of this approach to liver cancer was then carried out. An evolution-associated query signature related to the development and progression of liver cancer was first constructed for drug retrieval. Following the best practice approach, HHT was identified as the candidate agent for its highest reversal potency. Since the query signature ($Sig_{evo}$) could reflect the properties of liver cancer initiation and development, we considered that HHT might have both therapeutic and preventive effects on liver cancer. The therapeutic effect of HHT was assessed by in vitro cell line models as well as in vivo subcutaneous xenograft model. Both of them suggested remarkable tumor-killing activity of HHT. For validating the preventive effect, an indirect approach that focused on proving the antifibrotic effect of HHT was adopted. The results demonstrated that HHT could alleviate liver fibrosis in vivo and inhibit the activation of HSCs in vitro. Inhibition of liver fibrogenesis might prevent the progression of cirrhosis and thereby suppress HCC tumorigenesis (*Fuchs et al., 2014*). Therefore, we supposed that HHT had the potential to be taken as preventive agents for liver cancer as well. Notably, in view of the grim prognosis and imperfect treatment modalities of liver cancer, prevention of HCC development in patients at high risk of primary malignancy rather than treating patients at advanced stages is theoretically the most desirable approach to improve patient prognosis (*Fujiwara et al., 2018*; *Nakagawa et al., 2016*). As HHT has been approved by FDA for the treatment of chronic myelogenous leukemia, it can be tested directly in clinic without worrying about its safety problem (*Kantarjian et al., 2013*).

In this study, we have performed the most comprehensive surveys so far about the influencing factors of signature reversion-based drug prediction. Two novel benchmarking standards are proposed, providing new insight into the evaluation of related methodologies. All the findings in this study are verified independently by at least two different approaches, ensuring the reliability of the conclusions. Nevertheless, we also recognize several important limitations. First, with our design, our conclusions are conditional and hold only under the conditions of using compound profiles of HepG2 in LINCS as reference signatures. Further investigations using other LINCS data are required to extend current conclusions to other conditions. Second, the parameters recommended by us, including topN of 200 and query signature size of 100, are more or less based on our subjective judgments and should be taken as a rough guide. Although there are sufficient non-quantitative estimates supporting the use of these two parameters, more efforts are still needed to accurately determine the optimal parameters. Third, we only focused on analyzing the data from the project that utilized transcriptomic platforms to measure cell responses during perturbation experiments, and other omics data which are actively being generated by different LINCS centers might also be a good choice for computational drug discovery and repositioning (*Keenan et al., 2018*; *Koleti et al., 2018*). Recently, large-scale resources (CPPA) of perturbed protein responses have been generated (*Zhao et al., 2020*). Considering that proteins are the components of the basic functional units in biological pathways, investigating the optimal repositioning strategy based on proteomic resources may also have important implications.

In summary, our findings fill a knowledge gap in the area of LINCS-based computational repositioning. Through exploiting these findings, we also determined a promising anti-liver cancer agent HHT, of which the therapeutic and preventive effects have been validated experimentally.

## Materials and methods

**Key resources table**

| Reagent type (species) or resource | Designation | Source or reference | Identifiers | Additional information |
|---|---|---|---|---|
| Cell line (*Homo sapiens*) | Hep3B | ATCC | Cat#: HB-8064; RRID:CVCL_0326 | |
| Cell line (*H. sapiens*) | HepG2 | ATCC | Cat#: HB-8065; RRID:CVCL_0027 | |
| Cell line (*H. sapiens*) | Huh6 | RCB | Cat#: RCB1367; RRID:CVCL_4381 | |
| Cell line (*H. sapiens*) | Huh7 | JCRB | Cat#: JCRB0403; RRID:CVCL_0336 | |
| Cell line (*H. sapiens*) | MHCC97H | Zhongshan Hospital | RRID:CVCL_4972 | Liver Cancer Institute of Zhongshan Hospital (Shanghai, China) |
| Cell line (*H. sapiens*) | PLC/PRF/5 | ATCC | Cat#: CRL-802; RRID:CVCL_0485 | |
| Cell line (*H. sapiens*) | SNU398 | ATCC | Cat#: CRL-2233; RRID:CVCL_0077 | |
| Cell line (*H. sapiens*) | SNU449 | ATCC | Cat#: CRL-2234; RRID:CVCL_0454 | |
| Cell line (*H. sapiens*) | SNU475 | ATCC | Cat#: CRL-2236; RRID:CVCL_0497 | |
| Cell line (*H. sapiens*) | SK-Hep1 | ATCC | Cat#: HTB-52; RRID:CVCL_0525 | |
| Cell line (*H. sapiens*) | LX2 | ATCC | Cat#: SCC064; RRID:CVCL_5792 | |
| Chemical compound, drug | Homoharringtonine | Selleck Chemicals | S9015 | |
| Antibody | Anti-HSP90 (Mouse monoclonal) | Santa Cruz Biotechnology | Cat#: sc-13119; RRID:AB_675659 | WB (1:5000) |
| Antibody | Anti-α-SMA (Mouse monoclonal) | Sigma-Aldrich | Cat#: A5228; RRID:AB_262054 | WB (1:2000) IF (1:200) |
| Antibody | Anti-Collagen I (Rabbit polyclonal) | ProteinTech | Cat#: 14695-1-AP; RRID:AB_2082037 | WB (1:2000) IF (1:200) |
| Sequence-based reagent | ACTA2_F | This paper | PCR primer | 5'GACAATGGCTCTGGGCTCTGTAA3' |
| Sequence-based reagent | ACTA2_R | This paper | PCR primer | 5'CTGTGCTTCGTCACCCACGTA3' |
| Sequence-based reagent | COL1A1_F | This paper | PCR primer | 5'TCCTGGTCCTGCTGGCAAAGAA3' |
| Sequence-based reagent | COL1A1_R | This paper | PCR primer | 5'CACGCTGTCCAGCAATACCTTGA3' |
| Software, algorithm | R software, version 3.6.0 | https://cran.r-project.org/ | RRID:SCR_001905 | |
| Software, algorithm | ImageJ, version 1.53k | http://imagej.net/ | RRID:SCR_003070 | |
| Software, algorithm | Combenefit, version 2.02 | https://sourceforge.net/projects/combenefit/ | | |

### LINCS data source and processing

We downloaded the LINCS level 5 data (moderated Z-score) which comprises the differential expression signatures for nearly 20,000 unique compounds as well as meta-information of these signatures from Gene Expression Omnibus (GEO) database (Phase I: GSE92742, Phase II: GSE70138). Because this study only focused on analyzing compound signatures, those signatures induced by other perturbagens including gene knockdown (knockout) and gene overexpression were first excluded. The L1000 platform used by LINCS project only measures the expression level of 978 landmark genes, and the expression of remaining genes was based on imputation (*Subramanian et al., 2017*). This set of landmark genes is widely expressed in various cellular contexts and can well represent the full genome (*Subramanian et al., 2017*). Accordingly, we chose to use just the landmark genes (*Chen et al., 2017a*;

*Chen et al., 2017b*). To ensure the reliability, only high-quality signatures are designated for following analyses (is_gold=1). In addition, L1000 data was further filtered for only 6-hr treatment samples due to the most abundant experiments on HepG2 cell line in this time point. Besides, the previous study also showed that gene expression changes obtained at a late time point (such as 24 hr) might reflect secondary or even tertiary responses, and the mechanistic effects of compounds might not be correctly recorded at late time point (*De Wolf et al., 2016*). As for the perturbation concentration, we selected expression profiles measured at 10 µM considering that this relatively high concentration is often chosen for performing high-throughput small molecular screens, and also, there exist the most abundant experiments at this concentration. The similarity between compound pairs was calculated based on cosine similarity algorithm (*Cheng et al., 2013*). For visualizing the LINCS data in 2D space, we measured the cosine distance (1−cosine similarity) between signatures and utilized cosine distance matrix as input to perform t-SNE analysis (*Niepel et al., 2017*). We downloaded the MOA and clinical phase information of compounds in LINCS from the Drug Repurposing Hub (https://clue.io/repurposing) (*Corsello et al., 2017*). The basal expression data of LINCS cell lines was achieved from the Cancer Cell Line Encyclopedia (CCLE) project (https://portals.broadinstitute.org/ccle/) (*Ghandi et al., 2019*). The expression-based similarity between different cell lines or between cell lines and clinical samples was determined through using ranked-based Spearman correlation (*Chen et al., 2017b*).

## AUC-based benchmarking standard

In this study, we mainly focused exclusively on one cell line (HepG2) and one disease (liver cancer). Two benchmarking standards, namely AUC-based standard and KS statistic-based standard, were generated for evaluating the retrieval performance of disease-compound similarity metrics across different conditions. For establishing AUC-based standard, we collected the drug response data from multiple data sources. Compound $IC_{50}$s tested in HepG2 cell line was achieved from ChEMBL (version 27) (*Mendez et al., 2019*) and Liver Cancer Model Repository (LIMORE) (*Qiu et al., 2019*) data sets. Compounds among LINCS, ChEMBL, and LIMORE were mapped using compound name followed by manual inspection. Each experiment provided in ChEMBL was also manually checked to ensure the compliance with our requirement. Due to the redundancy of $IC_{50}$s, the median $IC_{50}$s of certain compounds among duplicates was used for representing the activity of this compound. We categorized the compounds into effective ($IC_{50} < 10$ µM) and ineffective groups ($IC_{50} \geq 10$ µM) according to a previous study (*Chen et al., 2017a*). The ability to distinguish between effective and ineffective compounds was taken as a measurement of retrieval performance of different similarity metrics (namely AUC value) (*Chen et al., 2017a*). Notably, some have argued that partial AUC (limiting false positive rate at 0.1/0.01) might be a more relevant statistic for actual application of drug repositioning (*Cheng et al., 2013*; *Cheng et al., 2014*). However, due to the limited size of our benchmarking data set, adopting partial AUC could result in loss of statistical precision. The statistical significance of AUCs was calculated through performing permutation test. Briefly, we randomly permuted the class labels of the feature vectors and created 10,000 permutations to form a distribution of 'random' AUCs. Then, the p value was determined according to the fraction of 'random' AUCs greater than or equal to the observed AUC (*Cheng et al., 2014*). For distinguishing, AUC used for evaluating drug retrieval performance was renamed as drug retrieval-associated AUC (DR-AUC), the higher values of which indicate better performance.

## KS statistic-based benchmarking standard

To avoid confusion, it should be noted that KS-based method was also used for calculating disease-compound similarity scores, and the specific details were described below. For generating the benchmarking data set required for the KS statistic-based standard, we systematically surveyed clinical trials involved in HCC treatment and compiled a set of potential HCC agents (clinicaltrials.gov). Preliminary retrieval yielded 1999 results, and after removing trials failing to fulfill our requirements, we obtained 254 potential therapeutic agents for HCC. The detailed retrieval process was presented in *Figure 3A*. To minimize potential selection bias, this process was performed independently by two investigators (CY and XH). Perturbagen-induced expression profiles of 27 agents among these tested in HepG2 cell line are available in LINCS data set. Based on the assumption that HCC-associated agents are more likely to reverse HCC signature than random agent combinations, the enrichment capabilities of different similarity metrics could be used to assess their repositioning performance. The calculation

of enrichment score (ES) of HCC agents was generally identical with that in gene set enrichment analysis (*Subramanian et al., 2005*). Considering that there was no need to account for the size of the agents set, we did not calculate the normalized enrichment score (NES) that might introduce additional randomization. To obtain a nominal p value, we created 10,000 permutations and recomputed the ES for each permutation to form a null distribution. The significance p of the observed ES was then determined relative to the null distribution. Notably, ES used in retrieval performance evaluation was renamed as drug retrieval-associated ES (DR-ES), the lower values of which represent better performance.

## Other pharmacogenomic data sets

In addition to ChEMBL and LIMORE, the drug response data of HepG2 cell line was also obtained from the Cancer Therapeutics Response Portal (CTRP) data set (CTRPv.2.0, released October 2015) (*Rees et al., 2016*). Considering that IC$_{50}$s were not provided, the available AUDRC values were used solely for demonstrating the correlation between reversal potency and drug efficacy in different conditions. The AUC values in CTRP range from 0 to 30, and similar to IC$_{50}$, lower values indicate increased sensitivity to treatment. To investigate the drug sensitivity of repositioning candidate across different HCC cell lines, we achieved the response data from the PRISM Repurposing data set (19Q4, released December 2019). Although IC$_{50}$s are also provided by PRISM as one of drug response metrics, the drug response data of HepG2 cell line is absent (similar situation also exists in the Genomics of Drug Sensitivity in Cancer data set). Therefore, drug response data in these data sets were not used for developing AUC-based benchmarking standard.

## Genetic dependency data

CRISPR dependency data were obtained from the 20Q1 dependency map (DepMap) portal, which contained dependencies estimated for nearly 20,000 protein-coding genes and 739 cell lines using the CERES algorithm (*Meyers et al., 2017*). CERES score (gene effect) was used to measure the dependency of the gene of interest in cell lines, and a lower CERES score indicates a higher likelihood that the gene is essential in cell growth and survival. Besides, data of dependency probability were also achieved. A probability of dependency of certain gene in certain cell lines greater than 0.5 represents that the gene can be considered essential in this cell line. Essential genes in liver cancer were defined as genes that were essential in all 22 liver cancer cell lines.

## RNA-sequencing data sets

We collected five RNA-sequencing (RNA-seq)-based HCC cohorts, including CHCC-HBV (*Gao et al., 2019*), LICA-FR (*Schulze et al., 2015*), LIRI-JP (*Fujimoto et al., 2016*), TCGA-LIHC (*Ally et al., 2017*), and GSE124535 (*Jiang et al., 2019*), representing 947 HCC patients derived from four geographically different origins. Of these, LICA, LIRI, and LIHC cohorts provided raw counts quantifying gene expression, which were transformed into transcripts per kilobase million (TPM) values for subsequent analyses (raw counts were only used for *edgeR*-based differential expression analysis) (*Li and Dewey, 2011*). CHCC and GSE124535 cohorts provided fragments per kilobase per million reads (FPKM) normalized data, which was also converted to TPM values. All TPM values were log2 transformed. In addition, the batch effects-normalized expression matrices of ~10,000 patients across 33 human cancers (TCGA Pan-Cancer) were downloaded from the UCSC Xena browser (http://xena.ucsc.edu/). The RNA-seq data of 29 normal tissues were downloaded from the Genotype-Tissue Expression (GTEx) project (https://gtexportal.org/home/). Ensembl GeneIDs were mapped to HGNC symbols using *biomaRt* package.

## Array data sets

Five microarray-based clinical cohorts, including E-TABM-36 (*Kim et al., 2012*), GSE14520 (*Roessler et al., 2012*), GSE54236 (*Villa et al., 2016*), GSE76427 (*Grinchuk et al., 2018*), and GSE84005, were included to construct and validate HCC-associated signatures. Raw microarray data generated from Affymetrix platforms were normalized using robust multi-array average (RMA) method in *Affy* package (*Gautier et al., 2004*), while Illumina platform-derived raw data were normalized using the robust spline normalization (RSN) method in *lumi* package (*Du et al., 2008*). In other cases, normalized data were directly downloaded for use. Three liver cancer development-associated cohorts, including

GSE89377, GSE6764 (*Wurmbach et al., 2007*), and GSE15654 (*Hoshida et al., 2013*), were also included for constructing the new signature, Sig$_{evo}$, which was then applied to query LINCS for finding potential therapeutics of liver cancer. The samples in GSE89377 and GSE6764 covered multiple stages of the development of liver cancer, with the ability to recapitulate the step-wise process of hepato-carcinogenesis and progression. As for GSE15654, this cohort contains the gene expression profiles of samples from 216 patients with early cirrhosis who were prospectively followed for a median of 10 years, which thus can be used to identify the relationship between gene expression and HCC occurrence (*Hoshida et al., 2013*). In addition, a liver fibrosis-associated clinical cohort (GSE84044) (*Wang et al., 2017*) and four experimental data sets, including two carbon tetrachloride (CCl$_4$)-treated mouse data sets (GSE27640 and GSE71379) (*Fuchs et al., 2014*; *Nakagawa et al., 2016*) and two diethyl-nitrosamine (DEN)-treated rat data sets (GSE19057 and GSE63726) (*Fuchs et al., 2014*; *Nakagawa et al., 2016*), were utilized to further assess the potential implication of Sig$_{evo}$. Mouse and rat genes were mapped to orthologous human genes using *biomaRt* package, and genes without known human homologous relationships were excluded.

## Clinical data

Among the clinical cohorts above, nine cohorts have corresponding follow-up information, including four RNA-seq cohorts (CHCC, LICA, LIRI, and LIHC), and five microarray cohorts (E-TABM-36, GSE14520, GSE54236, GSE76427, and GSE15654). For RNA-seq cohorts, the survival data of CHCC and LICA cohort were obtained from the supplementary files of reference (*Gao et al., 2019*; *Schulze et al., 2015*), data of LIRI cohort were achieved from the International Cancer Genome Consortium (ICGC) portal (https://dcc.icgc.org/), and data of LIHC cohort were achieved from TCGA Pan-Cancer Clinical Data Resource (TCGA-CDR) (*Liu et al., 2018*). For microarray cohorts, complete clinical data were accessed from either public database (GEO: https://www.ncbi.nlm.nih.gov/gds/; ArrayExpress: https://www.ebi.ac.uk/arrayexpress/) or the original authors. Notably, except GSE15654 which uses the occurrence of HCC as endpoint, other eight cohorts all take survival status as endpoint.

## Signature matching methods

The retrieval performance of six different matching methods, including XSum (*Cheng et al., 2014*), XCos (*Cheng et al., 2013*; *Cheng et al., 2014*), XCor (*Zhou et al., 2018*), XSpe (*Zhou et al., 2018*), KS test (*Lamb et al., 2006*), and the RGES (*Chen et al., 2017a*), was systematically benchmarked. Based on the consideration that small variations in expression changes may be noise without biological significance, the eXtreme methods only utilized top up- and downregulated genes in compound signatures for similarity score calculation (all remaining genes were assigned the values of zero). By contrast, KS and RGES methods use complete compound profiles as reference signatures. The detailed features and scoring schemes of these methods are described as follows.

The XSum method handles the up- and downregulated genes separately. In brief, the sums of the change values in reference/compound signatures relative to upregulated query/disease genes (sum$_{up}$) and downregulated query/disease genes (sum$_{down}$) are first calculated. Then, XSum is defined as following: XSum=sum$_{up}$−sum$_{down}$. Other three eXtreme methods, including XCos, XCor, and XSpe, take disease signatures as a whole to query compound signatures, and they calculated the correlation between the numeric vectors of disease and compound signatures using cosine similarity, Pearson correlation, and spearman correlation, respectively. Notably, cosine similarity is nearly identical with Pearson correlation except without centering vectors around the mean values. The KS method was adopted by the first CMap study and has been the most widely used method for connecting disease signatures to compound signatures (*Lamb et al., 2006*). Similar to XSum, KS method also need to seperate disease signatures into two gene sets, upregulated gene set and downregulated gene set, and ignores the magnitude of differential expression. Briefly, using complete compound profiles as reference, maximum deviation (MD)-based ES of upregulated gene set (es$_{up}$) and downregulated gene set (es$_{down}$) are first computed. If es$_{up}$ and es$_{down}$ have the same algebraic sign then KSscore=0, otherwise, KSscore=es$_{up}$−es$_{down}$. The RGES method is a recently proposed modification of the original KS method, which was demonstrated to perform better in drug response prediction than KS method (*Chen et al., 2017a*). In contrast to original KS method, RGES focuses on the reversal relation between the disease and agents, and RGES is defined as es$_{up}$−es$_{down}$ regardless of the sign of es$_{up}$ and es$_{down}$. In addition to the above six methods, there also exist many other methods, such as WSS/sscMap

(*Zhang and Gant, 2008*), TES (*Iorio et al., 2010*), ProbCmap (*Parkkinen and Kaski, 2014*), NFFinder (*Setoain et al., 2015*), and EMUDRA (*Zhou et al., 2018*), for calculating the similarity between disease and compound signatures. However, some of them are not accessible currently and some are developed based on the data of initial CMap data set (1309 compounds). Accordingly, we did not include these methods in our analyses.

## Generation of query signatures for performance evaluation

For evaluating retrieval performance of similarity metrics at different conditions, we prepared four HCC-associated gene signatures to query LINCS. Two of them, $Sig_{gastro}$ and $Sig_{NC}$, are achieved from previously published studies (*Chen et al., 2017a*; *Chen et al., 2017b*). Given that the development of these two signatures was mainly based on LIHC cohort, as a complement, the other two were generated from another RNA-seq cohort (LIRI) and a microarray cohort (GSE54236), respectively. The differentially expressed genes in LIRI cohort were computed using *edgeR* package (version 3.26.5) on raw count data (*McCarthy et al., 2012*). For microarray data, we used *limma* package (version 3.40.2) to conduct differential expression analysis on normalized data (*Ritchie et al., 2015*). The statistically significant differential genes in the above analyses were defined as adjusted $p<0.01$ and absolute $\log_2$ fold change (FC)>1. As a result, we obtained a 70-gene signature ($Sig_{LIRI}$) with 48 up- and 22 downregulated genes from LIRI and a 28-gene signature ($Sig_{GSE54236}$) with 22 up- and 6 downregulated genes from GSE54236, respectively, which could represent discordant expression pattern of HCC. The gene numbers in signatures created through differential expression analysis were much less than that in prognostic signatures (see section below). To make these two types of signatures comparable, we relaxed the significance threshold of differential genes to $p<0.01$ and $\log_2 FC>0.5$, and built two increased signatures which included 125 (LIRI) and 116 genes (GSE54236). These two increased signatures were also used to explore the potential influences of signature size.

## Construction of size-diversified query signatures

We adopted two independent approaches to explore whether the differences of query/disease signature size could affect subsequent drug retrieval. The first approach was based on iterating the threshold of fold change values, ranging from 0.1 to 0.1 to the maximum/minimum with an increment/decrement of 0.05, which could obtain a number of query signatures with varying signature size (duplicates were removed). As for the second approach, two increased signatures, 125-gene signature from LIRI and 116-gene signature from GSE54236, were taken as the basis for generating smaller-size testing signatures. Briefly, we randomly extracted testing signatures from complete signatures, with the size ranging from the minimum of 5 to the maximum of 124 or 115. To avoid bias, this process was repeated 1000 times to generate 1000 testing signatures for each signature size.

## Construction of query signatures representing different clinical phenotypes

To investigate whether the clinical phenotype of signature was potential factor affecting the retrieval performance, we developed two strategies, a forward strategy starting from generation of signatures with distinguishing clinical phenotypes to the evaluation of retrieval performance and a backward strategy starting from obtaining signature with the best performance to the comprehensive investigations of its clinical implication. For the first strategy, to compare with general HCC signature representing discordant expression pattern, two prognostic signatures based on LIRI and GSE54236 cohorts were constructed. We integrated survival data with expression data and performed Cox proportional hazards regression to assess association between overall survival and gene expression. The statistically significant prognostic genes were defined as $p<0.005$. A 133-gene prognostic signature with 117 poor- and 16 good-outcome genes was generated based on LIRI, while analysis on GSE54236 resulted in a 107-gene prognostic signature with 79 poor- and 28 good-outcome genes. Comparisons of drug retrieval performance between these two types of signatures were carried out subsequently. For the second strategy, taking 978 landmark genes as a basis, simple random sampling without replacement (SRSWOR) was performed to extract genes from landmarks for forming candidate signatures. The size of randomized signatures was set at 100 and the process of random sampling was repeated 10,000 times to obtain a collection of 10,000 randomized signatures. The DR-AUC and DR-ES values

were then calculated for each signature, and the optimal one was defined as the signature with the minimum of DR-AUC multiplying DR-ES.

## Generation of evolution-associated query signature

To find compounds with potential to prevent and treat liver cancer, we developed a hepatocarcinogenesis and progression-associated signature. GSE89377 cohort was utilized to build this signature while GSE6764 cohort was taken for external validation. To identify genes associated with developmental stages, RF model was constructed, taking stages as dependent variable. Variable importance was assessed with the mean decrease accuracy (MDA) measures for individual factors in RF model. Variables with positive MDA values are of high importance in predicting stages. In other words, these variables are more likely to be related with liver cancer development and progression (negative MDA values can be regarded as equivalent to zero importance with no predictive power). RF analysis was independently repeated 1000 times with 1000 trees growing each time, and genes with positive MDA values incorporated in more than 200 iterations were kept for subsequent analyses.

We next performed WGCNA to assign resultant genes into modules according to expression similarity and recognize the trajectories of gene expression during liver cancer development (*Langfelder and Horvath, 2008*). First, an appropriate soft threshold was estimated by using the *pickSoftThreshold* function in *WGCNA* package. Then, we constructed WGCNA network and detected gene expression modules using *blockwiseModules* function with a minimum module gene number of 50, soft thresholded power of 12, and a dendrogram cut height of 0.3. Genes without assignment to specific modules were assigned the color of gray. Module eigengenes (MEs) representing the first principal components (PC1) of each module were returned, and the module-trait relationship (MTR) analysis was conducted by calculating the correlation between MEs and developmental stages. The expression trend of each module across seven stages of HCC development was visualized through using mean PC1 values of samples in each stage to generate trend curves. According to the correlation coefficient of MTR analysis and the visualized expression trend of each module, two modules exhibiting the highest positive/negative correlation with developmental stages as well as showing gradually increasing or decreasing expression trends were selected. Subsequently, to explore the biological processes associated with genes in these two modules, we conducted hypergeometric test based on the hallmark gene sets (h.all.v7.0.symbols) downloaded from the Molecular Signatures Database (MSigDB) using *enricher* function in *clusterProfiler* package (*Liberzon et al., 2015*; *Yu et al., 2012*). The p values from the hypergeometric tests were adjusted for multiple comparison testing and an adjusted p value less than 0.05 was considered significantly enriched.

Genes in these two modules were mapped to the 978 landmark genes, resulting in a 159-gene panel (134 genes in ascending module and 25 genes in descending module). According to the findings described in Results section, we further narrowed down this panel to create a query signature with 100 genes. The molecular and clinical data in GSE15654 were utilized to determine the association between the expression patterns of signatures and the occurrence of HCC. Briefly, we first performed SRSWOR to extract a subset of 100 genes from the 159-gene panel, repeated 10,000 times. As a result, 10,000 randomized signatures with 100 genes per signature were generated. Next, PC1 values of all randomized signatures were extracted based on expression data from GSE15654 to represent the overall expression patterns of these signatures, and the follow-up data using HCC occurrence as endpoint was then integrated with above expression pattern data for subsequent Cox proportional hazards regression (COXPH). The signature which had the minimum p value across the 10,000 COXPH analyses was considered the optimal signature. The expression trend of this signature was further validated GSE6764 cohort.

## Human cell lines and compounds

The liver cancer cell lines, Hep3B, Huh7, PLC/PRF/5, SNU398, and Huh6, were provided by Erasmus University (Rotterdam, the Netherlands). MHCC97H and SK-Hep1 were provided by the Liver Cancer Institute of Zhongshan Hospital (Shanghai, China). SNU449, SNU475, HepG2, and the immortalized human HSC line LX2 were purchased from the American Type Culture Collection (ATCC). These cells were maintained in Dulbecco's modified Eagle's medium (DMEM) (Gibco, Carlsbad, CA) supplemented with 10% fetal bovine serum (FBS) (Gibco) and 1% penicillin/streptomycin (Basal Media), incubated at 37°C in humidified atmosphere with 5% $CO_2$. Mycoplasma contamination was excluded

via a PCR-based method. The identities of all the cell lines were confirmed by short tandem repeat (STR) profiling. Human recombinant transforming growth factor β1 (TGF-β1) was purchased from R&D Systems (Minneapolis, MN), which was used to activate LX2 (10 ng/ml TGF-β1 for 24 hr). HHT treatment was performed by pre-treating for 2 hr before TGF-β1 stimulation. HHT (S9015) was purchased from Selleck Chemicals and dissolved in dimethyl sulfoxide (DMSO) using a storage concentration of 10 mM.

## Cell proliferation assays

For long-term cell proliferation assay, cells were seeded into six-well plates ($2-3\times10^4$ cells per well) and HHT was added after 24 hr. Cells were treated with HHT as indicated for 10 days during which the culture media were replaced every 3 days. Then, cells were stained with 1% crystal violet for 10 min and rinsed with tap water. Pictures were taken using ImageScanner III (GE Healthcare) at 300-dpi resolution. For IncuCyte real-time assay, cells were cultured and seeded into 96-well plates at a density of 1000–1500 cells per well, and 24 hr later, HHT was added at indicated concentrations. Cells were imaged every 4 hr in IncuCyte ZOOM system (Essen Bioscience) and phase-contrast images were collected and analyzed to determine the proliferation curves based on cell confluence. Cell viability in dose-response matrix was assessed using CellTiter-Blue (CTB) assay (Promega) according to the manufacturer's recommendations. For measuring the synergistic effect of HHT-sorafenib combination, three different models, including Bliss independence model, Loewe additivity model and Highest single agent (HAS) model, were adopted, which were all implemented in Combenefit software version 2.02 (*Di Veroli et al., 2016*).

## Xenografts

Male BALB/c nude mice of 6–8 weeks old were used to establish xenograft tumor model. MHCC97H cells were suspended in 200-µl serum-free DMEM and subcutaneously injected into the upper flank of each mouse. When tumors reached a volume of approximately 50–100 mm³, mice from both groups were randomly assigned to treatment with vehicle or HHT (1 mg/kg, daily gavage). The 1 mg/kg dosage of HHT used to treat the nude mice with xenograft tumors was selected according to previous studies (*Wang et al., 2021*; *Weng et al., 2018*). Tumor volume was monitored every 3–4 days. The body weights were monitored every day. After 2 weeks of treatment, the mice were euthanized, the tumors were weighed and imaged.

## Liver fibrosis model

Mouse model of liver fibrosis was established based on previous publications (*Chen et al., 2014*; *Qu et al., 2018*; *Scholten et al., 2015*). In specific, 6-week-old male C57BL/6 mice (Shanghai Model Organisms Center, Shanghai, China) were treated three times a week for 10 weeks with intragastric administration of 0.1 ml of a 40% solution of CCl₄ (Aladdin, Shanghai, China) in olive oil (N=12) or olive oil alone (N=6). A subset of CCL4-treated mice received daily gavage of either 0.2 mg/kg HHT (N=6) or vehicle (N=6) during weeks 6–10. A concentration of 0.2 mg/kg was selected considering that the concentration for prevention is typically much lower than that for anti-tumor therapy (*Bayo et al., 2021*). Mice were sacrificed 3 days after the final treatment. The liver was harvested and cardiac terminal blood draw was also performed.

## Histology

Formalin-fixed samples were embedded in paraffin, cut into 5-µm-thick sections. Histologic slides were stained with hematoxylin and eosin (H&E), Masson's trichrome, and Sirius Red according to standard procedures, and then scanned using the Aperio CS Scanscope (Aperio Technologies, CA, USA). Fibrosis score was assessed on Masson's trichrome staining using Ishak scoring system (*Ishak, 1994*) and the positive area of Sirius Red staining was quantified by ImageJ software (version 1.53k, http://imagej.net/). All slides were reviewed in a blinded fashion by the same expert pathologist.

## Liver function tests

Blood was collected by a cardiac blood draw at the time of sacrifice. Blood was allowed to clot at least 20 min and serum was purified by centrifugation. Serum was stored at –80°C prior to use. Liver function was evaluated through measuring several serological markers, including ALT, AST, alkaline

phosphatase (ALP), and albumin (Alb). The serum levels of these markers were determined by BS-200 Chemistry Analyzer (Mindray, China).

## Quantitative real-time PCR

We first harvested cells using TRIzol reagent (Invitrogen) based on the manufacturer's instruction. Then, cDNA synthesis was carried out using Maxima Universal First Strand cDNA Synthesis Kit (No. K1661, Thermo Fisher Scientific). Quantitative reverse transcription PCR (qRT-PCR) assays were conducted using 7500 Fast Real-Time PCR System (Applied Biosystems). Relative mRNA levels of genes shown were normalized to the mRNA level of glyceraldehyde-3-phosphate dehydrogenase (GAPDH) (house-keeping gene). The primer sequences for assays using SYBR Green master mix (Roche) are as follows:

> β-actin Forward, 5'AAATCTGGCACCACACCTTC3',
> β-actin Reverse, 5'GGGGTGTTGAAGGTCTCAAA3',
> Collagen I Forward, 5'TCCTGGTCCTGCTGGCAAAGAA3',
> Collagen I Reverse, 5'CACGCTGTCCAGCAATACCTTGA3',
> α-SMA Forward, 5'GACAATGGCTCTGGGCTCTGTAA3',
> α-SMA Reverse, 5'CTGTGCTTCGTCACCCACGTA3'.

## Western blotting analysis

Cells were washed with PBS and lysed on ice with RIPA lysis buffer supplemented with Complete Protease Inhibitor (Roche) and Phosphatase Inhibitor Cocktails II and III (Sigma). Protein concentration was measured using the BCA Protein Assay Kit (Pierce). All lysates were then freshly prepared and processed with Novex NuPAGE Gel Electrophoresis Systems (Thermo Fisher Scientific) followed by western blotting. The antibody against α-smooth muscle actin (α-SMA) (A5228) was obtained from Sigma-Aldrich (USA) and the antibody against collagen I (14695-1-AP) was achieved from ProteinTech.

## Immunofluorescence

Cells were cultured on glass cover slips, fixed for 10 min with 4% formaldehyde, and permeabilized with 0.5% Triton X-100 for 15 min at room temperature. Immunofluorescence analysis was performed using the following antibodies: anti-Actin, α-Smooth Muscle antibody (1:200), anti-collagen I (1:200), anti-mouse IgG Fab2 Alexa Fluor (R) 488 (1:2000, CST), and anti-rabbit IgG Fab2 Alexa Fluor (R) 542 (1:2000, CST). Cell nuclei were stained with DAPI (4,6-diamidino-2-phenylindole). After immunostaining, the samples were observed using a LEICA TCS SP5 confocal microscope.

## RNA sequencing

For RNA sequencing, total RNA was extracted and purified using the TRIzol reagent (Invitrogen). The library was prepared using TruSeq RNA sample prep kit according to the manufacturer's protocol (Illumina). Paired-end libraries were sequenced by an Illumina HiSeq 4000, with a sequence coverage of 20 million paired reads. For data analysis, raw sequencing reads were mapped to the human genome (GRCh38) using STAR (version 2.4.2g1) (*Dobin et al., 2013*). Then gene-level read counts were generated using featureCounts from the subRead package with default settings (*Liao et al., 2014*).

## Statistics

All the computational analyses and graphical visualization were performed in R software, version 3.6.0 (https://cran.r-project.org/). Unless stated otherwise, correlation between two continuous variables was measured by Spearman's rank-order correlation, and pairwise comparisons were conducted using Kruskal-Wallis and Wilcoxon sum-rank tests. ROC curves and AUC values were visualized and calculated using the pROC package (*Robin et al., 2011*). The hazard ratio was estimated using Cox regression model in *survival* R package. Cumulative hazard curve was carried out using *jskm* package and the log-rank test was used to determine the statistical significance of differences. All data points indicate individual biologic replicates (independent experimental samples) and not technical replicates (the same sample re-analyzed using the same method). A two-tailed $p < 0.05$ was considered significant unless indicated otherwise.

## Acknowledgements

This work was presented in part as an oral presentation at the Cold Spring Harbor Asia Conference, held on December 7–10, 2021, Virtual Liver Meeting.

## Additional information

### Funding

| Funder | Grant reference number | Author |
|---|---|---|
| National Natural Science Foundation of China | 81972208 | Hui Wang |
| National Natural Science Foundation of China | 82170646 | Hualian Hang |
| Shanghai Natural Science Foundation | 19ZR1452700 | Hui Wang |
| The Interdisciplinary Program of Shanghai Jiao Tong University | YG2021ZD10 | Hualian Hang |

The funders had no role in study design, data collection and interpretation, or the decision to submit the work for publication.

### Author contributions

Chen Yang, Conceptualization, Data curation, Formal analysis, Investigation, Methodology, Software, Visualization, Writing - original draft, Writing - review and editing; Hailin Zhang, Linmeng Zhang, Formal analysis, Validation; Mengnuo Chen, Conceptualization, Formal analysis, Investigation, Methodology, Writing - original draft, Writing - review and editing; Siying Wang, Data curation, Formal analysis, Methodology, Resources, Visualization; Ruolan Qian, Formal analysis; Xiaowen Huang, Jun Wang, Zhicheng Liu, Resources; Wenxin Qin, Writing - review and editing; Cun Wang, Data curation, Project administration, Supervision, Writing - review and editing; Hualian Hang, Conceptualization, Formal analysis, Methodology, Writing - review and editing; Hui Wang, Conceptualization, Funding acquisition, Project administration, Resources, Supervision, Writing - review and editing

### Author ORCIDs

Chen Yang http://orcid.org/0000-0002-1237-2632
Cun Wang http://orcid.org/0000-0002-4977-2189
Hui Wang http://orcid.org/0000-0003-4947-9537

### Ethics

All animals were manipulated according to protocols approved by the Shanghai Medical Experimental Animal Care Commission and Shanghai Cancer Institute, Renji Hospital, Shanghai Jiao Tong University School of Medicine.

### Decision letter and Author response

Decision letter https://doi.org/10.7554/eLife.71880.sa1
Author response https://doi.org/10.7554/eLife.71880.sa2

## Additional files

### Supplementary files

- Supplementary file 1. Pharmacogenomic and transcriptomic datasets used in this study.
- Supplementary file 2. Cosine similarity within different drugs of LINCS.
- Supplementary file 3. Benchmarking dataset for AUC-based standard.
- Supplementary file 4. Benchmarking dataset for KS statistic-based standard.
- Supplementary file 5. A summary of four query signatures used for benchmarking signature matching methods.

- Supplementary file 6. Results of AUC/KS-based performance evaluation of six signature matching methods.
- Supplementary file 7. The list of genes in Sig(evo).
- Supplementary file 8. Results of computational drug repositioning for liver cancer.
- Transparent reporting form

## Data availability

Sequencing data have been deposited in GEO under accession codes GSE180243 and GSE193897. All data generated or analysed during this study are included in the manuscript and supporting files.

The following datasets were generated:

| Author(s) | Year | Dataset title | Dataset URL | Database and Identifier |
|---|---|---|---|---|
| Chen Y | 2021 | A survey of optimal strategy for signature-based drug repositioning and an application to liver cancer | https://www.ncbi.nlm.nih.gov/geo/query/acc.cgi?acc=GSE180243 | NCBI Gene Expression Omnibus, GSE180243 |
| Chen Y | 2022 | A survey of optimal strategy for signature-based drug repositioning and an application to liver cancer (liver cancer cell lines) | https://www.ncbi.nlm.nih.gov/geo/query/acc.cgi?acc=GSE193897 | NCBI Gene Expression Omnibus, GSE193897 |

The following previously published datasets were used:

| Author(s) | Year | Dataset title | Dataset URL | Database and Identifier |
|---|---|---|---|---|
| Aurélien de R, Charles B, David R, Dominique F, Emmanuelle J, Jacques B, Jean S, Jessica Z-R, Paulette B-S, Pierre L-P, Sandra R, Sandrine B | 2007 | Transcription profiling of 57 hepato cellular carcinoma tumoral samples, 3 hepatocellular adenomas, 5 non-tumoral pools | https://www.ebi.ac.uk/arrayexpress/experiments/E-TABM-36/ | ArrayExpress, E-TABM-36 |
| Jiang Y, Zhang L | 2019 | Gene expression profiles of 35 paired HCC and non-tumor tissues by RNA-seq data | https://www.ncbi.nlm.nih.gov/geo/query/acc.cgi?acc=GSE124535 | NCBI Gene Expression Omnibus, GSE124535 |
| Ally A, Balasundaram M, Carlsen R, Chuah E, Clarke A, Dhalla N | 2016 | Comprehensive and integrative genomic characterization of hepatocellular carcinoma | https://xenabrowser.net/datapages/?cohort=TCGA%20Liver%20Cancer%20(LIHC)&removeHub=https%3A%2F%2Fxena.treehouse.gi.ucsc.edu%3A443 | Xena Functional Genomics Explorer, TCGA Liver Cancer (LIHC) |
| Ghandi M, Huang FW, Jané-Valbuena J, Kryukov GV, Lo CC, McDonald ER | 2018 | Broad Institute Cancer Cell Line Encyclopedia (CCLE) | https://depmap.org/portal/download/ | DepMap Portal, CCLE |
| GTEx Consortium | 2017 | The Genotype-Tissue Expression (GTEx) project | https://gtexportal.org/home/ | Genotype-Tissue Expression (GTEx), GTEx |

*Continued on next page*

*Continued*

| Author(s) | Year | Dataset title | Dataset URL | Database and Identifier |
|---|---|---|---|---|
| Cancer Genome Atlas Research Network | 2016 | The Cancer Genome Atlas Pan-Cancer analysis project | https://xenabrowser. net/datapages/? cohort=TCGA% 20Pan-Cancer% 20(PANCAN)& removeHub=https% 3A%2F%2Fxena. treehouse.gi.ucsc. edu%3A443 | Xena Functional Genomics Explorer, TCGA Pan-Cancer (PANCAN) |
| Subramanian A, Narayan R, Corsello SM, Peck DD, Natoli TE, Lu X | 2017 | A Next Generation Connectivity Map: L1000 Platform and the First 1,000,000 Profiles | https://clue.io/GEO-guide | CMap LINCS Gene Expression Resource, LINCS |
| Mendez D, Gaulton A, Bento AP, Chambers J, De Veij M, Félix E | 2019 | ChEMBL: towards direct deposition of bioassay data | https://www.ebi.ac. uk/chembl/ | European Bioinformatics Institute, ChEMBL |
| Rees MG, Seashore-Ludlow B, Cheah JH, Adams DJ, Price EV, Gill S | 2016 | Cancer Therapeutics Response Portal | https://portals. broadinstitute.org/ ctrp/ | Broad Institute, CTRP |
| Qiu Z, Li H, Zhang Z, Zhu Z, He S, Wang X | 2019 | Liver Cancer Model Repository | https://www.picb.ac. cn/limore/batch | LIMORE, LIMORE |
| Corsello SM, Nagari RT, Spangler RD, Rossen J, Kocak M, Bryan JG | 2020 | PRISM Repurposing dataset | https://depmap.org/ repurposing/ | DepMap Portal, PRISM |
| Wang XW | 2010 | Gene expression data of human hepatocellular carcinoma (HCC) | https://www.ncbi. nlm.nih.gov/geo/ query/acc.cgi?acc= GSE14520 | NCBI Gene Expression Omnibus, GSE14520 |
| Villa E, Critelli R, Lei B, Marzocchi G, Cammà C, Giannelli G, Pontisso P, Colopi S, Caporali C, Cabibbo G, Milosa F, Maccio L, Martinez-Chantar ML, Todesca P, Turola E, Berselli A, De Maria N, Ballestri S, Schepis F, Loria P, Gerunda GE, Losi L, Di Benedetto F, Cillo U | 2014 | Prospective gene expression analysis of human RNA samples from Hepatocellular Carcinoma in relation with growth rate and survival | https://www.ncbi. nlm.nih.gov/geo/ query/acc.cgi?acc= GSE54236 | NCBI Gene Expression Omnibus, GSE54236 |
| Grinchuk OV, Yenamandra SP, Kuznetsov VA | 2017 | Microarray expression data for tumor and adjacent non-tumor tissues from hepatocellular carcinoma patients | https://www.ncbi. nlm.nih.gov/geo/ query/acc.cgi?acc= GSE76427 | NCBI Gene Expression Omnibus, GSE76427 |
| Tu X, Song J, Chen X, He F, Zhou G | 2017 | Integrative omics analysis in HCC samples [mRNA expression] | https://www.ncbi. nlm.nih.gov/geo/ query/acc.cgi?acc= GSE84005 | NCBI Gene Expression Omnibus, GSE84005 |

*Continued*

| Author(s) | Year | Dataset title | Dataset URL | Database and Identifier |
|---|---|---|---|---|
| Wurmbach E, Chen Y, Khitrov G, Zhang W, Roayaie S, Schwartz M, Fiel I, Thung S, Mazzaferro V, Bruix J, Bottinger E, Friedman S, Waxman S, Llovet JM | 2007 | Genome-wide molecular profiles of HCV-induced dysplasia and hepatocellular carcinoma | https://www.ncbi.nlm.nih.gov/geo/query/acc.cgi?acc=GSE6764 | NCBI Gene Expression Omnibus, GSE6764 |
| Hoshida Y, Villanueva A, Sangiovanni A, Sole M, Gould J, Gupta S, Taylor B, Crenshaw A, Gabriel S, Minguez B, Iavarone M, Friedman S, Colombo M, Llovet JM, Golub TR | 2013 | Gene-expression profiles of hepatitis C-related, early-stage liver cirrhosis | https://www.ncbi.nlm.nih.gov/geo/query/acc.cgi?acc=GSE15654 | NCBI Gene Expression Omnibus, GSE15654 |
| Eun J, Nam S | 2017 | Identifying novel drivers of human hepatocellular carcinoma and revealing clinical relevance as early diagnostic and prognostic biomarker | https://www.ncbi.nlm.nih.gov/geo/query/acc.cgi?acc=GSE89377 | NCBI Gene Expression Omnibus, GSE89377 |
| Wang M, Lu L, Zhang J, Yuan Z, Zhang X | 2016 | Characterization of gene expression profile in HBV-related liver fibrosis patients | https://www.ncbi.nlm.nih.gov/geo/query/acc.cgi?acc=GSE84044 | NCBI Gene Expression Omnibus, GSE84044 |
| Fuchs BC, Hoshida Y, Fujii T, Yamada S, Lauwers GY, McGinn CM, Wei L, Kuroda T, Lanuti M, Gupta S, Crenshaw A, Onofrio R, Taylor B, Winckler W, Golub TR, Tanabe KK | 2014 | Gene expression profile of liver tissue in carbon tetrachloride (CCl4)-treated mouse treated with erlotinib | https://www.ncbi.nlm.nih.gov/geo/query/acc.cgi?acc=GSE27640 | NCBI Gene Expression Omnibus, GSE27640 |
| Fuchs BC, Hoshida Y | 2016 | Gene expression profile of liver tissue from carbon tetrachloride (CCl4)-treated mouse cultured ex vivo | https://www.ncbi.nlm.nih.gov/geo/query/acc.cgi?acc=GSE71379 | NCBI Gene Expression Omnibus, GSE71379 |
| Fuchs BC, Hoshida Y, Fujii T, Lauwers GY, McGinn CM, Yamada S, Kuroda T, Lanuti M, Golub TR, Tanabe KK | 2014 | Gene expression profile of liver tissue in low-dose, repeated diethylnitrosamine (DEN)-treated rat treated with erlotinib | https://www.ncbi.nlm.nih.gov/geo/query/acc.cgi?acc=GSE19057 | NCBI Gene Expression Omnibus, GSE19057 |
| Fuchs BC, Hoshida Y | 2016 | Gene expression profiles of fractionated cells from cirrhotic rat livers | https://www.ncbi.nlm.nih.gov/geo/query/acc.cgi?acc=GSE63726 | NCBI Gene Expression Omnibus, GSE63726 |

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
