## [Editor Report]

This paper describes a new method and experimental manipulations to identify homoharringtonine as a new potential therapy for liver cancer and the underlying liver disease.

---

## [Decision Letter]

**Decision letter after peer review:**

Thank you for submitting your article "A survey of optimal strategy for signature-based drug repositioning and an application to liver cancer" for consideration by *eLife*. Your article has been reviewed by 2 peer reviewers, and the evaluation has been overseen by a Reviewing Editor and a Senior Editor. The reviewers have opted to remain anonymous.

As is customary in *eLife*, the reviewers have discussed their critiques with one another. What follows below is the Reviewing Editor's edited compilation of the essential and ancillary points provided by reviewers in their critiques and in their interaction post-review. Please submit a revised version that addresses these concerns directly. Although we expect that you will address these comments in your response letter, we also need to see the corresponding revision clearly marked in the text of the manuscript. Some of the reviewers' comments may seem to be simple queries or challenges that do not prompt revisions to the text. Please keep in mind, however, that readers may have the same perspective as the reviewers. Therefore, it is essential that you attempt to amend or expand the text to clarify the narrative accordingly.

Essential revisions:

1) The authors, should determine the HHT ic50 in normal cells to determine if this compound have cancer selectivity.

2) The authors identify the HHT compound due to its potential to reverse a gene signature on HepG2 cells at 10 uM. Additionally, the authors reported an IC50 for HHT near to 0.4 μm which is ~20 fold lower than the used to generate the transcriptomic data in the LINCS project. Therefore authors should validate the capability of HHT to reverse the gene signature in more than one cell line, and using a lower dose of the compound (~IC50).

3) The comparisons between the expression of RPL3 in T vs NT tissue was only shown in two data sets. This analysis should be expanded to other HCC datasets.

4) The potential of HHT as antitumoral and antifibrotic should be determined performing in vivo experiments.

---

## [Author Response]

Essential revisions:1) The authors, should determine the HHT ic50 in normal cells to determine if this compound have cancer selectivity.

According to this suggestion, the inhibitory effect of HHT on non-tumor cells was tested. Six non-tumor cell lines, including 4 liver-derived cell lines (L-02, WRL68, THLE-3, and HepaRG) and 2 non-liver-derived cell lines (BJ and RPE), were collected from ATCC or as gifts from other labs. Of these, L-02 and THLE-3 were not authenticated by short tandem repeat (STR) profiling and HepaRG was found to have mycoplasma contamination, and thus these three cell lines were excluded. Long-term cell proliferation assay was then performed on WRL68, BJ and RPE cell lines at the same HHT concentrations used in liver cancer cell lines. Disappointingly, we did not observe remarkable cancer-selective inhibitory effect of HHT (Author response image 1).

**Author response image 1. sa2fig1:** Validation of cancer-selective inhibitory effect of HHT. (A) Long-term cell proliferation assays of normal and liver cancer cell lines. (B) Quantitative results of long-term cell proliferation assays. Crystal violet was solubilized using 33% glacial acetic acid for 20 min and the absorbance was measured at 590 nm. The statistical significance of difference between groups was determined using Student’s *t* test. Data are represented as mean ± SD. (C) Body weight changes of mice in control (N = 6) and HHT-treated (N = 6) groups. The statistical significance was determined using Student’s t test. Data are represented as mean ± SD. (D) Comparison of serum levels of ALT, AST and ALB between control (n = 4) and HHT-treated (n = 4) groups. The statistical significance of difference between groups was determined using Student’s t test. Data are represented as mean ± SD.

Although above result did not meet our initial expectation, we speculated that the therapeutic application of HHT in liver cancer remains possible and our previous conclusion can still be valid. The reasons are as follows. First, a higher cancer selectivity indicates a lower off-tumor toxicity. For new drugs, safety evaluation is an essential part of the drug development. By contrast, for repurposed drugs, we can worry less about their safety problems, since most of them have well-established safety and tolerability profiles. HHT has been approved by FDA for the treatment of adult patients with chronic myeloid leukemia (CML) since 2012 (Alvandi et al., 2014), and a plethora of studies demonstrate that HHT can be adequately tolerated by patients who have end-stage cancer, with a low incidence of serious adverse events at its therapeutic concentrations (Cortes et al., 2015; Damlaj, Lipton, and Assouline, 2016; Hwang et al., 2021; Nijenhuis et al., 2016; Short et al., 2019). In other words, HHT administration is feasible and the toxicities are clinically manageable.

In addition, in vivo experiments for investigating the toxicity of HHT were also carried out. As body weight loss in mice can reflect the toxicity of the HHT towards normal tissues, we measured the body weights of HHT-treated mice every three days as an indication of in vivo toxicity. A concentration of 1mg/kg was selected for in vivo experiments, at which HHT could effectively inhibit the growth of subcutaneous xenograft tumors (see below for more details). No lethal events were observed at this dose among treated mice and the reduction of body weight of mice was less than 15% across the whole experimental period, suggesting that the in vivo toxicity of HHT was minimal and manageable (Christensen et al., 2014; Gijsen et al., 2010) (Author response image 1). Further, hepatotoxicity of HHT was also assessed by examining the levels of serum aspartate aminotransferase (AST), alanine aminotransferase (ALT), and albumin (ALB) at the end of the treatment period. No significant differences in these parameters between HHT-treated and control groups were observed, which implicated that HHT also had limited effect on the liver functions (Author response image 1).

Taken collectively, although we fail to demonstrate the selective anti-tumor activity of HHT in vitro, both in vivo and clinical evidence suggests that HHT treatment could be safe and well-tolerated, and thus the therapeutic application of HHT in liver cancer is still worthy of further consideration.

2) The authors identify the HHT compound due to its potential to reverse a gene signature on HepG2 cells at 10 uM. Additionally, the authors reported an IC50 for HHT near to 0.4 μm which is ~20 fold lower than the used to generate the transcriptomic data in the LINCS project. Therefore authors should validate the capability of HHT to reverse the gene signature in more than one cell line, and using a lower dose of the compound (~IC50).

We thank for this constructive suggestion. To investigate whether the reversal effect of HHT is cell type- or concertation-dependent, we generated HHT-perturbed expression data of five different liver cancer cell lines (Hep3B, HepG2, Huh6, Huh7 and PLC/PRF/5) and four different concentrations (0.1 μM, 0.5 μM, 1 μM and 10 μM). Of note, a fixed concentration 10 μM (a standard concentration in CMap and LINCS) was used to treat different cell lines and a single cell line HepG2 (a cell line used in LINCS) was perturbed by different concentrations (Author response image 2). RNA sequencing was performed to measure the gene expression, and then the HHT-induced expression signatures (HHT signatures) were obtained through calculating the fold changes relative to control groups. To validate the reversal effect of HHT on Sig_evo_ across different conditions, gene set enrichment analysis (GSEA) was conducted against different HHT signatures, taking ascending and descending genes in Sig_evo_ as query gene sets separately. The results showed that the ascending genes in Sig_evo_ tended to enrich in HHT-induced down-regulated genes (enrichment score < 0) (Figure 7—figure supplement 1B–C), while descending genes in Sig_evo_ appeared to be more associated with HHT-induced up-regulated genes (enrichment score > 0) (Figure 7—figure supplement 1B–C), suggesting that the ability of HHT to reverse the Sig_evo_ was independent of cell type and treatment concentration. All the data generated in this revision has been uploaded to the GEO database (accession number: GSE193897).

**Author response image 2. sa2fig2:** Schematic figure illustrating the experimental design. The expression profiles of 26 samples (16 HHT-treated and 10 control samples) were generated by RNA sequencing.

Besides, we also validated the reversal effect of HHT using previously generated perturbed expression profiles of LX-2 cells. It could be observed that HHT also had a certain capacity to reverse the expression of Sig_evo_ in LX-2 cells, further corroborating above conclusion (Author response image 3).

**Author response image 3. sa2fig3:** Reversal effect of HHT on LX-2 cells. A positive ES of a given gene set indicates the enrichment of HHT-induced up-regulated genes and a negative ES indicates the enrichment of HHT-induced down-regulated genes.

3) The comparisons between the expression of RPL3 in T vs NT tissue was only shown in two data sets. This analysis should be expanded to other HCC datasets.

We apologize for our omission that only partial results were presented in the previous submission. Comprehensive comparisons of the expression of RPL3 between tumor and non-tumor tissues have been conducted using seven liver cancer clinical cohorts with available expression profiles of tumor and non-tumor tissues in the revision (GSE84005 was excluded due to the lack of corresponding data of RPL3) (Author response image 4 and Figure 7—figure supplement 2A). RPL3 exhibits higher expression levels in tumor compared with non-tumor tissues in more than half the clinical cohorts (57.1%), demonstrating the plausibility of taking RPL3 as a potential target for treating liver cancer. Notably, RPL3 was observed to have higher expression levels in non-tumor tissues in GSE76427, which might attribute to the quality of the dataset itself, since another critical liver cancer biomarker AFP was also found to have abnormal expression pattern in this dataset (data not shown).

**Author response image 4. sa2fig4:** Comparisons of RPL3 expression between tumor and non-tumor tissues across seven liver cancer clinical cohorts. Data are presented as median ± quartiles, N ≥ 100. The statistical significance of difference between groups was determined using Wilcoxon sum rank tests.

4) The potential of HHT as antitumoral and antifibrotic should be determined performing in vivo experiments.

According to this suggestion, in vivo experiments for validating the antitumor as well as antifibrotic activities of HHT were carried out. MHCC97H cell line was selected for the subcutaneous xenograft mouse model. The dosage of HHT used to treat the nude mice with xenograft tumors (1mg/kg) was selected according to previous studies (Wang et al., 2021; Weng et al., 2018; Wolff et al., 2015). As expected, the result demonstrated that HHT could significantly inhibit the growth of xenograft tumors, with limited drug-related toxicity (Figure 7G–I).

The antifibrotic effect of HHT was validated using carbon tetrachloride (CCL_4_)-induced mouse liver fibrosis model (Chen et al., 2014; Qu et al., 2018). Based on a previous report, six-week CCL_4_ treatment is sufficient for fibrosis induction (Scholten, 2015). HHT administration was thus scheduled at week six and last for additional four weeks. To determine a nontoxic dose of HHT on C57BL/6 mice, we conducted a pilot experiment in 12 healthy male C57BL/6 mice treated with vehicle or HHT at varying concentrations (three mice in vehicle group, three mice in 0.2mg/kg HHT group, three mice in 0.5mg/kg HHT group and three mice in 1mg/kg HHT group). The result showed that HHT treatment at three concentrations all had limited effect on the animal weight (Author response image 5). Considering that the concentration for prevention is typically much lower than that for anti-tumor therapy, we ultimately selected the concentration of 0.2mg/kg for further experiments (Bayo et al., 2021) (Figure 8A). The degree of liver fibrosis was evaluated using Masson’s trichrome staining and Sirius Red staining. In addition, several serological indicators, including ALT, AST, ALB, and alkaline phosphate (ALP), were also measured to assess the liver function. The results suggested that HHT could impede fibrosis development and reduce liver injure in CCL_4_-induced mouse model, in agreement with our conjecture (Figure 8B–D).

**Author response image 5. sa2fig5:** Body weight changes of C57BL/6 mice receiving different treatments. Twelve mice were stratified into four groups, including vehicle group (N = 3), 0.2mg/kg HHT group (N = 3), 0.5mg/kg HHT group (N = 3) and 1mg/kg HHT group (N = 3). Data are represented as mean ± SD.

References

Alvandi, F., Kwitkowski, V. E., Ko, C. W., Rothmann, M. D., Ricci, S., Saber, H., . . . Pazdur, R. (2014). U.S. Food and Drug Administration approval summary: omacetaxine mepesuccinate as treatment for chronic myeloid leukemia. Oncologist, 19(1), 94-99. doi:10.1634/theoncologist.2013-0077

Bayo, J., Fiore, E. J., Dominguez, L. M., Cantero, M. J., Ciarlantini, M. S., Malvicini, M., . . . Mazzolini, G. D. (2021). Bioinformatic analysis of RHO family of GTPases identifies RAC1 pharmacological inhibition as a new therapeutic strategy for hepatocellular carcinoma. Gut, 70(7), 1362-1374. doi:10.1136/gutjnl-2020-321454

Chen, X., Gan, Y., Li, W., Su, J., Zhang, Y., Huang, Y., . . . Shi, Y. (2014). The interaction between mesenchymal stem cells and steroids during inflammation. Cell Death & Disease, 5(1), e1009. doi:10.1038/cddis.2013.537

Christensen, C. L., Kwiatkowski, N., Abraham, B. J., Carretero, J., Al-Shahrour, F., Zhang, T., . . . Wong, K. K. (2014). Targeting transcriptional addictions in small cell lung cancer with a covalent CDK7 inhibitor. Cancer Cell, 26(6), 909-922. doi:10.1016/j.ccell.2014.10.019

Cortes, J. E., Kantarjian, H. M., Rea, D., Wetzler, M., Lipton, J. H., Akard, L., . . . Nicolini, F. E. (2015). Final analysis of the efficacy and safety of omacetaxine mepesuccinate in patients with chronic- or accelerated-phase chronic myeloid leukemia: Results with 24 months of follow-up. Cancer, 121(10), 1637-1644. doi:10.1002/cncr.29240

Damlaj, M., Lipton, J. H., & Assouline, S. E. (2016). A safety evaluation of omacetaxine mepesuccinate for the treatment of chronic myeloid leukemia. Expert Opinion on Drug Safety, 15(9), 1279-1286. doi:10.1080/14740338.2016.1207760

Gijsen, M., King, P., Perera, T., Parker, P. J., Harris, A. L., Larijani, B., & Kong, A. (2010). HER2 phosphorylation is maintained by a PKB negative feedback loop in response to anti-HER2 herceptin in breast cancer. PLoS Biology, 8(12), e1000563. doi:10.1371/journal.pbio.1000563

Hwang, J., Singh, N., Braniecki, M., Gok Yavuz, B., Tsoukas, M. M., & Quigley, J. G. (2021). Omacetaxine added to a standard acute myeloid leukaemia chemotherapy regimen reduces cellular FLIP levels, markedly increasing the incidence of eccrine hidradenitis. British Journal of Haematology, 195(3), e138-e141. doi:10.1111/bjh.17715

Nijenhuis, C. M., Hellriegel, E., Beijnen, J. H., Hershock, D., Huitema, A. D., Lucas, L., . . . Schellens, J. H. (2016). Pharmacokinetics and excretion of (14)C-omacetaxine in patients with advanced solid tumors. Investigational New Drugs, 34(5), 565-574. doi:10.1007/s10637-016-0360-9

Qu, C., Zheng, D., Li, S., Liu, Y., Lidofsky, A., Holmes, J. A., . . . Hong, J. (2018). Tyrosine kinase SYK is a potential therapeutic target for liver fibrosis. Hepatology, 68(3), 1125-1139. doi:10.1002/hep.29881

Scholten, D. (2015). The carbon tetrachloride model in mice. Laboratory Animals, 49(1 Suppl), 4-11. doi:10.1177/0023677215571192

Short, N. J., Jabbour, E., Naqvi, K., Patel, A., Ning, J., Sasaki, K., . . . Garcia-Manero, G. (2019). A phase II study of omacetaxine mepesuccinate for patients with higher-risk myelodysplastic syndrome and chronic myelomonocytic leukemia after failure of hypomethylating agents. American Journal of Hematology, 94(1), 74-79. doi:10.1002/ajh.25318

Wang, H., Wang, R., Huang, D., Li, S., Gao, B., Kang, Z., . . . Yan, J. (2021). Homoharringtonine Exerts Anti-tumor Effects in Hepatocellular Carcinoma Through Activation of the Hippo Pathway. Frontiers in Pharmacology, 12, 592071. doi:10.3389/fphar.2021.592071

Weng, T. Y., Wu, H. F., Li, C. Y., Hung, Y. H., Chang, Y. W., Chen, Y. L., . . . Lai, M. D. (2018). Homoharringtonine induced immune alteration for an Efficient Anti-tumor Response in Mouse Models of Non-small Cell Lung Adenocarcinoma Expressing Kras Mutation. Scientific Reports, 8(1), 8216. doi:10.1038/s41598-018-26454-w

Wolff, N. C., Pavía-Jiménez, A., Tcheuyap, V. T., Alexander, S., Vishwanath, M., Christie, A., . . . Brugarolas, J. (2015). High-throughput simultaneous screen and counterscreen identifies homoharringtonine as synthetic lethal with von Hippel-Lindau loss in renal cell carcinoma. Oncotarget, 6(19), 16951-16962. doi:10.18632/oncotarget.4773